# Optimizing Noise Distributions for Differential Privacy

**Atefeh Gilani** [1]  **Juan Felipe Gomez** [2]  **Shahab Asoodeh** [3]  **Flavio P. Calmon** [4]  **Oliver Kosut** [1]  **Lalitha Sankar** [1]

## Abstract

We propose a unified optimization framework for designing continuous and discrete noise distributions that ensure differential privacy (DP) by minimizing Rényi DP, a variant of DP, under a cost constraint. Rényi DP has the advantage that by considering different values of the Rényi parameter $\alpha$, we can tailor our optimization for any number of compositions. To solve the optimization problem, we reduce it to a finite-dimensional convex formulation and perform preconditioned gradient descent. The resulting noise distributions are then compared to their Gaussian and Laplace counterparts. Numerical results demonstrate that our optimized distributions are consistently better, with significant improvements in $(\varepsilon, \delta)$-DP guarantees in the moderate composition regimes, compared to Gaussian and Laplace distributions with the same variance.

## 1. Introduction

Differential privacy (DP) (Dwork et al., 2006a;b) provides strong privacy protections by ensuring that information released from queries on sensitive data does not reveal whether any individual is included in the dataset. This is achieved through privacy mechanisms that add noise to query responses, effectively masking any private information.

In practice, privacy mechanisms are rarely applied only once to sensitive data. Instead, they are often used sequentially, with the frequency depending on the specific use case. In what we call the *moderate composition regime*, privacy mechanisms are applied a limited number of times to release aggregated data. For example, the U.S. Census Bureau

(Abowd et al., 2022) publishes demographic statistics such as population density or employment rates for each state, requiring repeated applications of privacy-preserving methods. On the other hand, the *large composition regime* involves thousands of sequential applications of privacy mechanisms, such as during the training of a machine learning model.

The most widely used noise distributions incorporated into privacy mechanisms for achieving DP are Laplace noise (Dwork et al., 2006b), typically employed when pure DP is desired, and Gaussian noise (Abadi et al., 2016), often used for approximate DP (i.e., $(\varepsilon, \delta)$-DP). However, these distributions may not necessarily be optimal across all settings in which they are applied. For example, while the Gaussian distribution is commonly used to ensure approximate DP in the large composition regime, the parametrized *Cactus distribution* has been shown to be optimal in this regime for a fixed sensitivity $s > 0$ (Alghamdi et al., 2022). Moreover, the *Schrödinger distribution* has been demonstrated to be optimal for vanishing sensitivity ($s \rightarrow 0^+$), and it only recovers the Gaussian distribution as a special case under a quadratic cost function (Alghamdi et al., 2023). Additionally, in (Geng et al., 2015), it was shown that the *Staircase distribution* is optimal for $\varepsilon$-DP in the single composition regime. In this paper, we focus on designing optimal noise distributions tailored to a fixed number of compositions and sensitivity. We demonstrate that the resulting optimal *Rényi DP noise distribution* significantly outperforms both Laplace and Gaussian distributions in the moderate composition regime.

Discrete noise distributions also hold significant value for DP. This is because finite precision implementations that generate continuous random variables can result in naive floating-point approximations which have been shown to compromise their *de facto* privacy guarantees (Mironov, 2012). Restricting noise distributions to integer support offers resilience against floating-point attacks and better suits applications involving integer-valued queries, such as population counts in the US Census. To address these challenges, the discrete Gaussian distribution (Canonne et al., 2020) and the discrete Laplace distribution (Ghosh et al., 2012; Balcer & Vadhan, 2017) have been proposed. These distributions, supported on integer values, serve as natural discrete analogues to the continuous Gaussian and Laplace distributions.

---

[1]School of Electrical, Computer and Energy Engineering, Arizona State University, Tempe, AZ, USA [2]Department of Physics, Harvard University, Cambridge, MA, USA [3]Department of Computing and Software, McMaster University, Hamilton, Ontario, Canada [4]School of Engineering and Applied Sciences, Harvard University, Cambridge, MA, USA. Correspondence to: Atefeh Gilani <agilani2@asu.edu>.

*Proceedings of the $42^{nd}$ International Conference on Machine Learning*, Vancouver, Canada. PMLR 267, 2025. Copyright 2025 by the author(s).

In this paper, we apply exactly the same approach as for continuous distributions to optimize discrete noise distributions. The comparison between the discrete Rényi DP noise distribution and the discrete Gaussian and Laplace distributions mirrors that of their continuous counterparts, with the discrete Rényi DP noise distribution also outperforming the discrete Gaussian and Laplace distributions in the moderate composition regime.

Whether in the continuous or discrete case, optimizing with respect to $(\varepsilon, \delta)$-DP in the context of compositions is challenging due to the problem's non-convex nature. To overcome this, we leverage the properties of Rényi differential privacy (RDP) (Mironov, 2017), a variant of DP defined in terms of the Rényi divergence of order $\alpha$, where $\alpha \in (1, \infty)$. RDP has garnered significant attention (Abadi et al., 2016; Chen et al., 2019; Feldman et al., 2021; Lécuyer et al., 2021; Feldman et al., 2022). One of its key advantages is that, for composition, the total RDP is the sum of the individual RDPs. Therefore, our approach is to optimize each RDP term individually, which is sufficient to optimize the overall privacy guarantee. In addition, RDP can be converted into $(\varepsilon, \delta)$-DP, as in the moments accountant in (Abadi et al., 2016; Mironov, 2017; Bun & Steinke, 2016; Balle et al., 2020; Asoodeh et al., 2021). This conversion involves minimizing over the Rényi order $\alpha$. Thus, the optimal $\alpha$ is closely tied to the number of compositions, and is generally decreasing with the number of compositions: it tends toward 1 in the large composition regime, and towards infinity in the single-composition regime. For moderate compositions, $\alpha$ generally lies within an intermediate range, avoiding the extremes of 1 and infinity.

Because our framework allows us to optimize the noise for any $\alpha$, our approach encompasses optimal noise distributions in both the large and small composition regimes. In the large composition regime, the Cactus distribution is shown to be optimal in (Alghamdi et al., 2022); this distribution is found by minimizing the Kullback-Leibler divergence, which corresponds to RDP as $\alpha \to 1$. Since we incorporate the moments accountant into our optimization framework, for large numbers of compositions, we will automatically seek $\alpha$ near 1, thus recovering the Cactus distribution. In the single-composition regime for pure DP, the optimal $\alpha$ tends to infinity, and our noise distribution recovers the Staircase distribution, consistent with RDP converging to pure DP as $\alpha \to \infty$.

Our main contributions include:

1. We propose an algorithm for optimizing noise distributions under $(\varepsilon, \delta)$-DP, outlined in Algorithm 1 [1]. Our procedure receives as input user-defined (i) target $\delta$, (ii) number of compositions, (iii) sensitivity, and (iv) average distortion per query (i.e., a cost constraint) such as mean

---

[1] Our implementation is available on GitHub (git, 2025).

---

**Algorithm 1** Optimal Noise Distribution for $(\varepsilon, \delta)$-DP

**input** : Privacy parameter $\delta$, Number of compositions $N_c$, Noise scale $\sigma$, Query sensitivity $s$, `type` (Discrete or Continuous), Hyperparameters $\theta$
**output** : Optimal distribution $P^*$ minimizing $\varepsilon$ for the given $\delta$ and $N_c$, satisfying $\mathbb{E}[\text{Cost}] \leq \sigma^2$
1: $P_0 \leftarrow InitializeDistribution(\text{type}, \sigma, s, \theta)$
2: $P^* \leftarrow OptimizationAlgorithm(P_0, \text{type}, \sigma, s, \delta, N_c, \theta)$

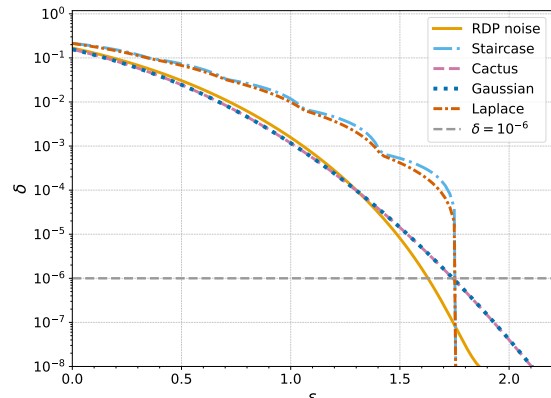

*Figure 1.* This plot compares the privacy curve of our optimized noise—designed for 10 compositions, $s = 1$, and $\delta = 10^{-6}$—against other noise distributions, all with the same standard deviation of 8. At the target $\delta = 10^{-6}$, our noise achieves an $\varepsilon$ of 1.62, compared to 1.76 for Laplace/Staircase and 1.74 for Gaussian/Cactus. Replacing these mechanisms with our optimized noise results in at least a 6.89% improvement in the $\varepsilon$ value

squared-error (MSE). The algorithm outputs an optimized discrete or continuous additive noise distribution. Figure 1 illustrates the resulting $(\varepsilon, \delta)$ curve for an optimized noise distribution considering 10 compositions, noise standard deviation constraint of 8, and target $\delta = 10^{-6}$. Relative to other noise mechanisms, our optimized mechanisms achieve a lower $\varepsilon$ for the same distortion at the target $\delta$.

2. We formulate a finite-dimensional convex optimization problem for producing optimized noise distributions and solve it using preconditioned gradient descent, denoted as *OptimizationAlgorithm* in Algorithm 1. This convex problem leverages RDP of order $\alpha$ as an intermediate optimization objective. The RDP hyperparameter $\alpha$ is automatically selected based on the the user-defined input parameters.

3. Our algorithm recovers as special cases noise distributions that are known to be optimal in different regimes, such as the the Staircase and Cactus mechanisms. In the moderate composition regime (roughly 10–40 compositions), both discrete and continuous distributions derived from our framework achieve superior $(\varepsilon, \delta)$-DP guarantees compared to their Gaussian and Laplace counterparts for a target cost,

number of compositions, and $\delta$. We validate the favorable performance of our optimized noise distributions using Connect-the-Dots (Doroshenko et al., 2022) accounting.

## 2. Preliminaries

*Notation.* Vectors are represented by bold lowercase letters, e.g., $\mathbf{x}$, and matrices by bold capital letters, e.g., $\mathbf{A}$. The symbol $\mathbf{B}^+$ denotes the Moore-Penrose right inverse of $\mathbf{B}$ (Wikipedia, a). The notation $\mathrm{diag}(\mathbf{x})$ denotes a diagonal matrix with the elements of $\mathbf{x}$. The symbol $[N]$ denotes the set of integers from 0 to $N$, i.e., $[N] = \{0, 1, 2, \ldots, N\}$. The symbol $\Phi$ represents the cumulative distribution function of the standard normal distribution.

We review some basic definitions and results from the DP literature. Given an alphabet $\mathcal{X}$, let $\mathcal{P}(\mathcal{X})$ be the set of distributions supported on $\mathcal{X}$. For $P, Q \in \mathcal{P}(\mathcal{X})$, the Rényi divergence of order $\alpha$, for $\alpha \in (0, 1) \cup (1, \infty)$, is

$$D_\alpha(P\|Q) = \frac{1}{\alpha - 1} \log \mathbb{E}_P \left( \frac{P(X)}{Q(X)} \right)^{\alpha - 1}. \quad (1)$$

Let $\mathcal{D}$ be a set of possible datasets, and $\sim$ be a "neighboring" relation among elements of $\mathcal{D}$. That is, for $d, d' \in \mathcal{D}$ we write $d \sim d'$ to mean that $d$ and $d'$ are neighbors, which typically means that they differ in one entry. A *mechanism* is a function $\mathcal{M} : \mathcal{D} \to \mathcal{P}(\mathcal{X})$, which, for each $d \in \mathcal{D}$, selects a distribution $\mathcal{M}_d \in \mathcal{P}(\mathcal{X})$. This can be interpreted as a conditional distribution for a random variable supported in $\mathcal{X}$ given a dataset from $\mathcal{D}$.

**Definition 2.1.** (Dwork et al., 2006a;b) A mechanism $\mathcal{M} : \mathcal{D} \to \mathcal{P}(\mathcal{X})$ is said to be $(\varepsilon, \delta)$-DP if for all $A \subset \mathcal{X}$,

$$\mathcal{M}_d(A) \leq e^\varepsilon \mathcal{M}_{d'}(A) + \delta \text{ for all } d, d' \in \mathcal{D}, \ d \sim d'. \quad (2)$$

For a mechanism $\mathcal{M}$, we also define the best $\varepsilon$ for a given $\delta$ as $\varepsilon_\mathcal{M}(\delta) = \inf\{\varepsilon : \mathcal{M} \text{ is } (\varepsilon, \delta)\text{-DP}\}$.

**Definition 2.2.** (Mironov, 2017) A mechanism $\mathcal{M} : \mathcal{D} \to \mathcal{P}(\mathcal{X})$ is said to be $(\alpha, \gamma)$-RDP if

$$D_\alpha(\mathcal{M}_d\|\mathcal{M}_{d'}) \leq \gamma \text{ for all } d, d' \in \mathcal{D}, \ d \sim d'. \quad (3)$$

For a mechanism $\mathcal{M}$, also define the best $\gamma$ for a given $\alpha$ as

$$\gamma_\mathcal{M}(\alpha) = \inf\{\gamma : \mathcal{M} \text{ is } (\alpha, \gamma)\text{-RDP}\}. \quad (4)$$

For two mechanisms $\mathcal{M}^{(1)}, \mathcal{M}^{(2)}$, each outputting a variable in $\mathcal{X}$, their *composition* $\mathcal{M} : \mathcal{D} \to \mathcal{P}(\mathcal{X} \times \mathcal{X})$ is

$$\mathcal{M}_d(x_1, x_2) = \mathcal{M}_d^{(1)}(x_1)\mathcal{M}_d^{(2)}(x_2). \quad (5)$$

**Proposition 2.3.** (Mironov, 2017) *For any mechanisms $\mathcal{M}^{(1)}, \mathcal{M}^{(2)}$ and their composition $\mathcal{M}$,*

$$\gamma_\mathcal{M}(\alpha) \leq \gamma_{\mathcal{M}^{(1)}}(\alpha) + \gamma_{\mathcal{M}^{(2)}}(\alpha). \quad (6)$$

The above is non-adaptive composition, in that each mechanism works independently of the other's output. In contrast, in an adaptive composition, the second mechanism may depend on the output of the first. A similar composition result holds for the adaptive setting (Mironov, 2017), but for brevity we omit the details. A particular consequence of Proposition 2.3 is that, if the same (or equivalent) mechanisms are composed $N_c$ times, then the RDP is simply multiplied by $N_c$.

The moments accountant (Abadi et al., 2016) provides a method to derive $(\varepsilon, \delta)$ guarantees from $(\alpha, \gamma)$ guarantees. The most basic form of the moments accountant follows.

**Proposition 2.4.** (Abadi et al., 2016) *For a mechanism $\mathcal{M}$,*

$$\varepsilon_\mathcal{M}(\delta) \leq \inf_{\alpha > 1} \gamma_\mathcal{M}(\alpha) + \frac{\log(1/\delta)}{\alpha - 1}. \quad (7)$$

While improvements to the moments accountant have been made in (Balle et al., 2020; Asoodeh et al., 2021), we use this simple version in our algorithm to simultaneously optimize for the noise distribution and $\alpha$. However, when we perform numerical privacy accounting for the resulting distribution, we use the state-of-the-art Connect-the-Dots accountant.

The *sensitivity* of a query $q : \mathcal{D} \to \mathbb{R}$ describes the maximum difference in the query's output when changing a single entry in the input dataset. Formally, it is defined as

$$s = \max_{d, d' \in \mathcal{D}, d \sim d'} |q(d) - q(d')|. \quad (8)$$

Given a real-valued query function $q$ with sensitivity bound $s$, the Gaussian mechanism involves adding Gaussian noise with zero mean and variance $\sigma^2$ to the query $q$. Similarly, the Laplace mechanism involves adding noise from a Laplace distribution with zero mean and scale parameter $t$ to the query $q$.

For many natural queries, the output is inherently discrete, specifically integer-valued, such as when counting how many records in a dataset meet a certain condition. In these cases, it is preferable to add discrete noise directly to the query results. Adding continuous noise to discrete results would require rounding, which can affect the privacy guarantees. Discrete Gaussian and discrete Laplace distributions are the discrete counterparts of their continuous versions and are specifically designed for discrete-valued queries. The definitions of discrete Gaussian and Laplace distributions are as follows. Let $\mathbb{Z}$ be the set of integers. The *discrete Gaussian distribution*, denoted $\mathcal{N}_\mathbb{Z}(\mu, \sigma^2)$, for $\mu \in \mathbb{Z}$ and $\sigma > 0$, is the PMF in $\mathcal{P}(\mathbb{Z})$ given by

$$P(x) = \frac{e^{-\frac{(x-\mu)^2}{2\sigma^2}}}{\sum_{y \in \mathbb{Z}} e^{-\frac{(y-\mu)^2}{2\sigma^2}}}, \quad x \in \mathbb{Z}. \quad (9)$$

The *discrete Laplace distribution*, denoted $\text{Lap}_{\mathbb{Z}}(\mu, t)$, for $\mu \in \mathbb{Z}$ and $t > 0$, is the PMF in $\mathcal{P}(\mathbb{Z})$ given by

$$P(x) = \frac{e^{1/t} - 1}{e^{1/t} + 1} \, e^{-|x-\mu|/t}, \quad x \in \mathbb{Z}. \tag{10}$$

## 3. Optimized Rényi DP Distributions

In this section, we find noise distributions with the best Rényi DP guarantees by formulating a minimax optimization problem. To set up this problem, we first give the following definitions. We use $\mathcal{Z}$ to denote the domain of the DP query output, which can be either the real numbers or the integers. The set $\mathcal{S}$ is defined as the intersection of the interval $[-s, s]$ with $\mathcal{Z}$, where $s \in \mathcal{Z}^+$ represents the query's sensitivity. Specifically, $\mathcal{S} = [-s, s]$ if $\mathcal{Z} = \mathbb{R}$, and $\mathcal{S} = \{-s, \ldots, 0, \ldots, s\}$ if $\mathcal{Z} = \mathbb{Z}$. We assume a cost function $c : \mathcal{Z} \to [0, \infty)$ that is symmetric, and an upper bound $C \in \mathbb{R}^+$ on the expected cost. For a function $f$, $T_t f$ is the shifted version of $f$ by $t$, i.e., $(T_t f)(x) := f(x - t)$.

With the above definitions, the optimization problem is

$$\underset{P_Z \in \mathcal{P}(\mathcal{Z})}{\text{minimize}} \; \max_{t \in \mathcal{S}} D_\alpha \left( P_Z \| T_t P_Z \right),$$
$$\text{subject to} \quad \mathbb{E}[c(Z)] \leq C. \tag{11}$$

For $\alpha > 1$, we can move the min-max operation inside the logarithm of the Rényi divergence, considering it as our main objective. Let $g_\alpha(P_Z, t)$ denote the expression inside the logarithm of $D_\alpha \left( P_Z \| T_t P_Z \right)$, defined as:

$$g_\alpha(P_Z, t) = \mathbb{E}_{P_Z} \left( \frac{P_Z}{T_t P_Z} \right)^{\alpha - 1}, \tag{12}$$

and let $g_\alpha(P_Z)$ denote the inner maximization:

$$g_\alpha(P_Z) = \max_{t \in \mathcal{S}} g_\alpha(P_Z, t). \tag{13}$$

We first show that $g_\alpha(P_Z)$ is convex in $P_Z$. By leveraging this property, along with the symmetry of the cost function, we restrict our search to symmetric noise distributions, as formalized in the following theorem. A detailed proof is provided in the Appendix A.

**Theorem 3.1.** *The function $g_\alpha(P_Z)$ is convex in $P_Z$. Moreover, it suffices to restrict the search to the set of symmetric noise distributions within $\mathcal{P}(\mathcal{Z})$ to solve the optimization problem in* (11).

### 3.1. Finite-Dimensional Distribution Classes

While we would like to solve (11), it cannot be solved as-is because the search space of distributions is infinite-dimensional. In this subsection, we address this by defining finite-dimensional families of distributions that can closely

approximate solutions to (11) while being computationally tractable. These families draw inspiration from a similar finite-dimensional parameterization in (Alghamdi et al., 2022). We then present a method for solving the optimization problem within these defined families.

For the continuous case, we focus on symmetric piecewise constant probability density functions (PDFs). Importantly, this approach is highly flexible, as any PDF can be closely approximated by using sufficiently small bin widths. This shifts the problem from optimizing the density at every real number to determining the probability $p_i$ of bin $i$. Given these probabilities, the PDF is given by

$$f(z) := \frac{p_{|i|}}{\Delta}, \quad \text{for } z \in I_i, \; i \in \mathbb{Z}, \tag{14}$$

where $I_i = \left( (i - \frac{1}{2})\Delta, (i + \frac{1}{2})\Delta \right)$ denotes the open interval corresponding to bin $i$, and $\Delta > 0$ represents the bin width. *Remark* 3.2. As the Lebesgue measure of the breakpoints in the piecewise-constant representation is zero, it is not necessary to explicitly define the value of $f$ at these points.

This formulation still involves countably infinite variables, specifically $\{p_i\}_{i \in \mathbb{Z}_{\geq 0}}$. To ensure a finite number of variables, we introduce the constraint that the distribution exhibits geometric tails. The geometric tails begin beyond a specific interval, starting after bin $N$, where $N$ is a hyperparameter for the distribution family. The decay rate of the geometric tail is given by a second hyperparameter $r$. Importantly, this tail assumption does not significantly impact the optimization, provided that the interval is sufficiently large to capture the majority of the PDF's significant behavior. For a fixed $\Delta$, these tails are fully characterized by the probability mass of the bins immediately preceding them and the decay factor $r$. This reduces the problem to determining the probability masses of the bins preceding the geometric tails, thus converting the problem into a finite-dimensional one. The following formally defines the family of distributions under consideration.

**Definition 3.3** (Symmetric Piecewise-Constant PDF Family). Let $N \in \mathbb{N}$, $r \in (0, 1)$, and $\Delta > 0$. The PDF associated with a vector of probabilities $\mathbf{p} = (p_0, p_1, \ldots, p_N) \in [0, 1]^{N+1}$ is

$$f_{\mathbf{p}, r, N, \Delta}(z) := \begin{cases} \dfrac{p_{|i|}}{\Delta}, & \text{if } z \in I_i, \; |i| < N, \\ \dfrac{p_N}{\Delta} r^{|i| - N}, & \text{if } z \in I_i, \; |i| \geq N, \end{cases} \tag{15}$$

subject to the normalization constraint

$$\int_{\mathbb{R}} f_{\mathbf{p}, r, N, \Delta}(z) \, dz = p_0 + 2 \sum_{j=1}^{N-1} p_j + \frac{2p_N}{1 - r} = 1. \tag{16}$$

For the discrete case, it is again sufficient to restrict our search to symmetric distributions. We also assume geomet-

ric tails as in the continuous setting to reduce to a finite-dimensional search space. The following gives the family of discrete distributions under consideration.

**Definition 3.4** (Symmetric PMF Family). Let $N \in \mathbb{N}$, $r \in (0, 1)$. The PMF associated with a vector of probabilities $\mathbf{p} = (p_0, p_1, \cdots, p_N) \in [0, 1]^{N+1}$ is

$$P_{\mathbf{p},r,N}(i) = \begin{cases} p_{|i|}, & \text{for } i \in \mathbb{Z}, \text{with } |i| \leq N \\ p_N r^{|i|-N}, & \text{for } i \in \mathbb{Z}, \text{with } |i| > N, \end{cases} \quad (17)$$

subject to the normalization constraint:

$$\sum_{i \in \mathbb{Z}} P_{\mathbf{p},r,N}(i) = p_0 + 2 \sum_{i=1}^{N-1} p_i + \frac{2p_N}{1-r} = 1. \quad (18)$$

The following proposition characterizes the expected cost for these two distribution families.

**Proposition 3.5.** *Let $Z$ be a random variable with distribution given by either the continuous or discrete distribution families described above. Then the expected cost is*

$$\mathbb{E}[c(Z)] = p_0 A_0 + 2 \sum_{i=1}^{N-1} p_i A_i + 2p_N \sum_{i=N}^{\infty} r^{i-N} A_i, \quad (19)$$

*where, for a continuous distribution,*

$$A_i = \frac{1}{\Delta} \int_{z \in I_i} c(z) \, dz, \quad \forall i \in \mathbb{Z}_{\geq 0}, \quad (20)$$

*and for a discrete distribution $A_i = c(i)$. In the special case where $c(z) = z^2$, corresponding to the variance[2] of $Z$, the expression simplifies to, for the continuous case,*

$$\text{Var}(Z) = \frac{\Delta^2}{12} + 2\Delta^2 \sum_{i=1}^{N-1} p_i \, i^2$$
$$+ 2p_N \Delta^2 \frac{r^2(N-1)^2 + N^2(1-2r) + r(2N+1)}{(1-r)^3}. \quad (21)$$

*For the discrete case, the variance is the same as in (21), except that the $\Delta^2/12$ term is not present, and $\Delta = 1$.*

The proof is provided in Appendix B.

### 3.2. Finite-Dimensional Convex Optimization

Within the distribution family described in Definition 3.3 or Definition 3.4, the task of finding the optimal distribution in (11) reduces to determining the vector $\mathbf{p} = (p_0, \ldots, p_N)$. Since $g_\alpha(P_Z)$, defined in (13), is convex in $P_Z$, it is also convex in $\mathbf{p}$. Therefore, the minimization component of the optimization reduces to a finite-dimensional convex optimization problem. The following theorem characterizes

---

[2] The distribution's symmetry ensures $Z$ has a mean of zero.

$g_\alpha(P_Z, t)$ in (12) for the two distribution families, and delineates the feasible regions for both continuous and discrete cases. Notably, the resulting finite-dimensional convex optimization problem is almost the same for the two domains.

**Theorem 3.6.** *Let $s$ be the sensitivity of a query. Within the symmetric piecewise-constant PDF family defined in Definition 3.3, with a specified bin width of $\Delta$ such that $\frac{s}{\Delta}$ is an integer, the optimization problem for the continuous case, as detailed in (11), can be reformulated as follows:*

$$\underset{\mathbf{p}=(p_0,p_1,\cdots,p_N)}{\text{minimize}} \; \max_{t \in \left\{1,\ldots,\frac{s}{\Delta}\right\}} g_\alpha(\mathbf{p}, t)$$

$$\text{subject to} \quad \mathbb{E}[c(Z)] \leq C,$$
$$p_0 + 2 \sum_{j=1}^{N-1} p_j + \frac{2\,p_N}{1-r} = 1,$$
$$p_i \in [0, 1] \quad \text{for all } i \in [N], \quad (22)$$

*where*

$$g_\alpha(\mathbf{p}, t) = \frac{p_N\, r}{1-r} \left( r^{(1-\alpha)t} + r^{\alpha t} \right) + \sum_{j=-N}^{N-t} p_{|t+j|}^{\alpha}\, p_{|j|}^{1-\alpha}$$

$$+ p_N^\alpha\, r^{\alpha(t-N)} \sum_{j=N-t+1}^{N} r^{\alpha j}\, p_{|j|}^{1-\alpha}$$

$$+ p_N^{1-\alpha}\, r^{-N(1-\alpha)} \sum_{j=-t-N}^{-N-1} p_{|t+j|}^{\alpha}\, r^{-j(1-\alpha)}. \quad (23)$$

*For the discrete case within the symmetric PMF family from Definition 3.4, the optimization problem in (11) can also be reformulated as (22)–(23) except that we take $\Delta = 1$. The cost constraint $\mathbb{E}[c(Z)] \leq C$ for both cases is further detailed in Proposition 3.5.*

*Proof Sketch:* The objective in (22) is the quantity inside the logarithm of the Rényi divergence — thus the RDP can be easily calculated from the optimal objective. Although the inner maximization in the continuous case initially considers $t \in [-s, s]$, we demonstrate that the divergence is maximized when the bins of a piecewise constant PDF align with those of its shifted version. In this scenario, the optimal shift is a multiple of the bin width $\Delta$ within the interval $[-s, s]$. The set $\{1, \ldots, \frac{s}{\Delta}\}$ corresponds to the possible integer multiples of $\Delta$ that determine the optimal shift values. The detailed proof is in Appendix C.

In the piecewise-constant PDF family introduced in Definition 3.3, setting $\Delta = 1$ creates a PDF where each bin has a width of 1 and is centered on the integers, which can be interpreted as a PMF over the integers. This leads to the objective for the discrete case being a specific instance of the continuous case when $\Delta = 1$.

Theorem 3.6 highlights that solving for the optimal noise in the two domains (continuous and discrete) is nearly identical.

**Algorithm 2** Initialize Distribution

---

**input** : Noise scale $\sigma$, `type` (discrete or continuous), Query sensitivity $s$, Distribution parameters $N, r, \Delta$

**output** : $\mathbf{p}_0$

1: $\hat{C}_{\min} \leftarrow 0$
2: $\hat{C}_{\max} \leftarrow 2\sigma^2$
3: **repeat**
4: $\quad \hat{C} \leftarrow \dfrac{\hat{C}_{\min} + \hat{C}_{\max}}{2}$
5: $\quad$ **for** $i = 0$ **to** $N - 1$ **do**
6: $\quad\quad p_i \leftarrow \Phi\big((i + \tfrac{1}{2})\Delta/\sqrt{\hat{C}}\big) - \Phi\big((i - \tfrac{1}{2})\Delta/\sqrt{\hat{C}}\big)$
7: $\quad$ **end for**
8: $\quad p_N \leftarrow (1 - r)\big[1 - \Phi\big((N - \tfrac{1}{2})\Delta/\sqrt{\hat{C}}\big)\big]$
9: $\quad$ Var $\leftarrow$ variance of the noise associated with $\mathbf{p}$ and `type` using Proposition 3.5
10: $\quad$ **if** Var $< \sigma^2$ **then**
11: $\quad\quad \hat{C}_{\min} \leftarrow \hat{C}$
12: $\quad$ **else**
13: $\quad\quad \hat{C}_{\max} \leftarrow \hat{C}$
14: $\quad$ **end if**
15: **until** Var and $\sigma^2$ are sufficiently close
16: $\mathbf{p}_0 \leftarrow \mathbf{p}$

---

**Algorithm 3** Optimization Algorithm

---

**input** : Privacy parameter $\delta$, Number of compositions $N_c$, Noise scale $\sigma$, Query sensitivity $s$, Total number of iterations $K$, Initial distribution $\mathbf{p}_0$, Distribution parameters $N, r, \Delta$, Rényi parameter $(\alpha)$ update time step $T$, `type` (discrete or continuous)

**output** : Final solution $\mathbf{p}_K$

1: $\alpha \leftarrow \sqrt{\dfrac{2\log(1/\delta)}{N_c}}\dfrac{\sigma}{s} + 1$
2: Calculate $\mathbf{A}, \mathbf{b}$ so the linear constraints for cost and normalization are given by $\mathbf{A}\mathbf{p} = \mathbf{b}$
3: **for** $k = 1$ **to** $K$ **do**
4: $\quad t^* \leftarrow \arg\max_{t \in \{1, \cdots, \frac{s}{\Delta}\}} g_\alpha(\mathbf{p}_{k-1}, t)$
5: $\quad \mathbf{M} \leftarrow \mathrm{diag}(\mathbf{p}_{k-1})^{-1}$
6: $\quad \mathbf{g} \leftarrow \mathbf{M}^{-1}\nabla_\mathbf{p} g_\alpha(\mathbf{p}_{k-1}, t^*)$
7: $\quad \mathbf{B} \leftarrow \mathbf{A}\mathbf{M}^{-1}$
8: $\quad \mathbf{B}^+ \leftarrow \mathbf{B}^T(\mathbf{B}\mathbf{B}^T)^{-1}$
9: $\quad \mathbf{g}^{\mathrm{proj}} \leftarrow \mathbf{g} - \mathbf{B}^+\mathbf{B}\mathbf{g}$
10: $\quad \mu^{\mathrm{ub}} \leftarrow \min_{i \in [N], \mathbf{g}_i^{\mathrm{proj}} > 0} \frac{1}{\mathbf{g}_i^{\mathrm{proj}}}$
11: $\quad \mathbf{p}_k \leftarrow \mathbf{p}_{k-1}$
12: $\quad g_{\min} \leftarrow g_\alpha(\mathbf{p}_{k-1}, t^*)$
13: $\quad$ **for** $\mu \in \left\{\mu^{\mathrm{ub}}, \frac{\mu^{\mathrm{ub}}}{2}, \ldots, \frac{\mu^{\mathrm{ub}}}{2^{10}}\right\}$ **do**
14: $\quad\quad \hat{\mathbf{p}} \leftarrow \mathbf{M}^{-1}(\mathbb{1} - \mu\, \mathbf{g}^{\mathrm{proj}})$
15: $\quad\quad \hat{g} \leftarrow \max_{t \in \{1, \cdots, \frac{s}{\Delta}\}} g_\alpha(\hat{\mathbf{p}}, t)$
16: $\quad\quad$ **if** $\hat{g} < g_{\min}$ **then**
17: $\quad\quad\quad \mathbf{p}_k \leftarrow \hat{\mathbf{p}}$
18: $\quad\quad$ **end if**
19: $\quad$ **end for**
20: $\quad$ **if** $k$ is a multiple of $T$ **then**
21: $\quad\quad \alpha \leftarrow \alpha - \dfrac{\gamma'_{\mathbf{p}_k, N_c, \delta}(\alpha)}{\gamma''_{\mathbf{p}_k, N_c, \delta}(\alpha)}$
22: $\quad$ **end if**
23: **end for**

---

The objectives differ only in that $\Delta$ must be 1 for the discrete case. Moreover, as shown in Proposition 3.5, for quadratic cost, the constraints differ only in that the continuous case includes an extra constant term $\Delta^2/12$ whereas discrete does not.

*Remark* 3.7. Let $\sigma$ represent the standard deviation of the noise. As demonstrated in (Mironov, 2017), for Gaussian and Laplace distributions, the RDP of order $\alpha$ depends solely on the ratio $\sigma/s$, rather than on $\sigma$ and $s$ individually. In fact, our noise distributions satisfy exactly the same property. In the optimization problem in (22), for a quadratic cost with $C = \sigma^2$, we can rewrite the cost constraint from (21) as:

$$
\frac{\sigma^2}{\Delta^2} = \frac{1}{12} + 2\sum_{i=1}^{N-1} p_i\, i^2
$$
$$
+ 2p_N \frac{r^2(N-1)^2 + N^2(1-2r) + r(2N+1)}{(1-r)^3}. \quad (24)
$$

Moreover, $s$ appears only in the set over which we maximize. By fixing $s/\Delta$ to an integer, say $m$, which is feasible because $\Delta$ can be adjusted to maintain the desired ratio, we get $\Delta = s/m$. Substituting this into (24), it becomes a function of $\sigma/s$. This implies that the optimization now depends solely on the ratio $\sigma/s$. Therefore, the RDP of our noise depend solely on the ratio $\sigma/s$, similar to the behavior observed for Gaussian and Laplace distributions.

## 4. Preconditioned Gradient Descent Algorithm

In this section, we present our algorithms for identifying the noise distribution that achieves the best $(\varepsilon, \delta)$-DP guarantees. Algorithm 1 is the overall algorithm, which calls Algorithm 2 and then Algorithm 3. The role of Algorithm 2 is to initialize the vector $\mathbf{p}$, which is passed to Algorithm 3. This latter algorithm performs preconditioned gradient descent to find the optimal noise distribution.

**Algorithm 2 Overview**: Since the Gaussian distribution is a straightforward baseline, we initialize with an approximation of this distribution. Since our distribution families cannot exactly capture the Gaussian (or discrete Gaussian) distribution, we must slightly adjust the parameters to exactly match the desired $\sigma$. In the algorithm, $\hat{C}$ represents the variance of the Gaussian: based on this, we derive $\mathbf{p}$ to approximate a Gaussian with variance $\hat{C}$. However, the resulting distribution will have a slightly different variance, denoted Var. We employ a bisection algorithm to find $\hat{C}$ so

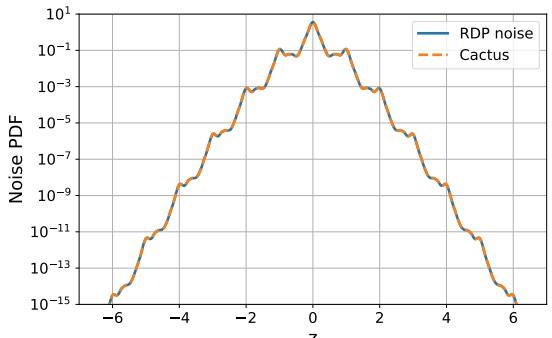
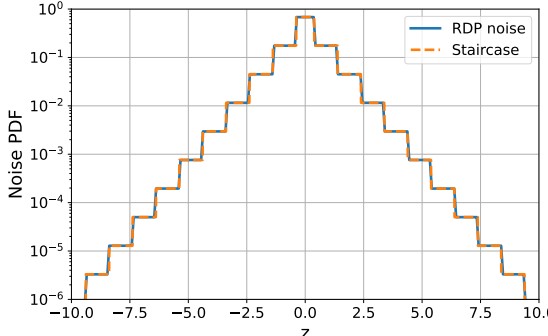

*Figure 2.* The left plot compares our optimized noise distribution and a Cactus distribution, both with standard deviation 0.3, under the setting $\delta = 10^{-5}$, $s = 1$, and 50,000 compositions, corresponding to $\alpha = 1.006$ (moments accountant), with noise parameters $\Delta = 0.005$, $N = 1600$, and $r = 0.9999$. The right plot compares our optimized noise and a Staircase distribution, both with standard deviation 1, under the setting $\delta = 10^{-20}$, $s = 1$, and one composition, corresponding to $\alpha = 125.01$, with noise parameters $\Delta = 0.01$, $N = 2000$, and $r = 0.9999$.

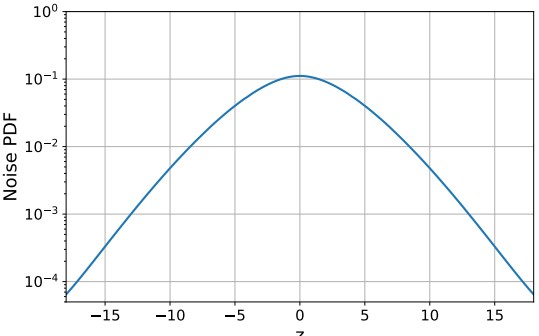
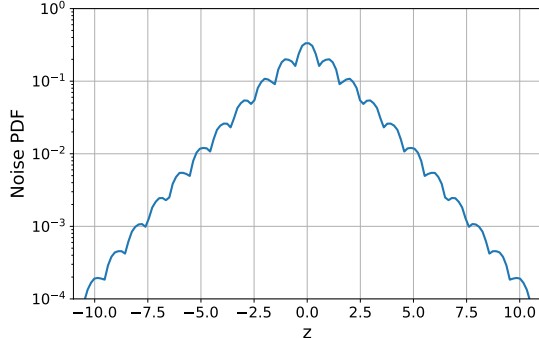

*Figure 3.* The left plot displays the optimized noise distribution for a standard deviation of 4, $\delta = 10^{-6}$, $s = 1$, and 20 compositions, corresponding to $\alpha = 5.95$ (using the moments accountant). The right plot shows the distribution for a standard deviation of 2, $\delta = 10^{-10}$, $s = 1$, and 8 compositions, corresponding to $\alpha = 11.32$. The noise parameters are $\Delta = 0.01$, $r = 0.9999$, and $N = 8000$ for the left plot, and $N = 4000$ for the right.

that Var $= \sigma^2$. The bisection method allows us to converge to the desired variance for the initial noise mechanism iteratively. It does so by iteratively tightening the chosen upper and lower bounds on $\hat{C}$ until they converge.

**Algorithm 3 Overview:** The main task is to solve the optimization problem described from Theorem 3.6, given by (22). The minimization problem is convex in **p**, so gradient descent will convergence to the optimal objective value. Recall that our optimization is a minimax problem. The maximization in (22) only requires iterating over a finite set, while a preconditioned gradient descent method is used to solve the convex optimization part.

**Explanation of Key Aspects of Algorithm 3:**

**Linear constraints:** The optimization problem involves two linear constraints, corresponding to the cost constraint (which is active) and the normalization constraint. These two constraints can be expressed as a matrix equation: $\mathbf{Ap} = \mathbf{b}$. We focus on the quadratic cost $c(z) = z^2$ with

$C = \sigma^2$, where $\sigma$ is the standard deviation of the noise. The appropriate cost constraint (depending on whether `type` is discrete or continuous) from Proposition 3.5 is used in each case. The normalization constraint is in (22).

**Optimization for $\alpha$:** Algorithm 3 simultaneously optimizes for **p** and $\alpha$. Specifically, we optimize $\alpha$ according to the moments accountant formula, given by

$$\gamma_{\mathbf{p}, N_c, \delta}(\alpha) = \frac{N_c}{\alpha - 1} \log g_\alpha(\mathbf{p}) + \frac{\log(1/\delta)}{\alpha - 1}, \qquad (25)$$

where $g_\alpha(\mathbf{p}) = \max_{t \in \{1, \ldots, s/\Delta\}} g_\alpha(\mathbf{p}, t)$. The initial value of $\alpha$ is set to the optimal $\alpha$ for the moments accountant with Gaussian noise (corresponding to the initialization of **p** to being approximately Gaussian). Using the fact that the RDP of Gaussian noise is $\frac{s^2 \alpha}{2\sigma^2}$ in the moments accountant formula, the optimal $\alpha$ for Gaussian is $\alpha^* = \sqrt{\frac{2 \log(1/\delta)}{N_c}} \frac{\sigma}{s} + 1$.

Subsequently, every $T$ iterations, $\alpha$ is updated by taking a Newton step to optimize the moments accountant formula.

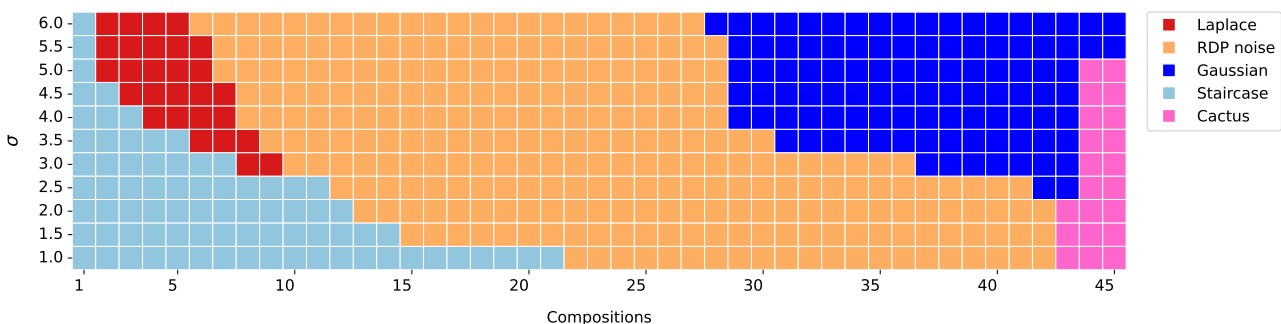

*Figure 4.* For fixed $\delta = 10^{-6}$, the grid evaluates different combinations of composition count and noise standard deviation ($\sigma$), marking RDP noise as the winner whenever it achieves more than a 2% improvement in $\varepsilon$ over the best alternative (among Gaussian, Laplace, Staircase, and Cactus). Even when another distribution is marked as the best, our noise still consistently outperforms the others, although by less than 2%.

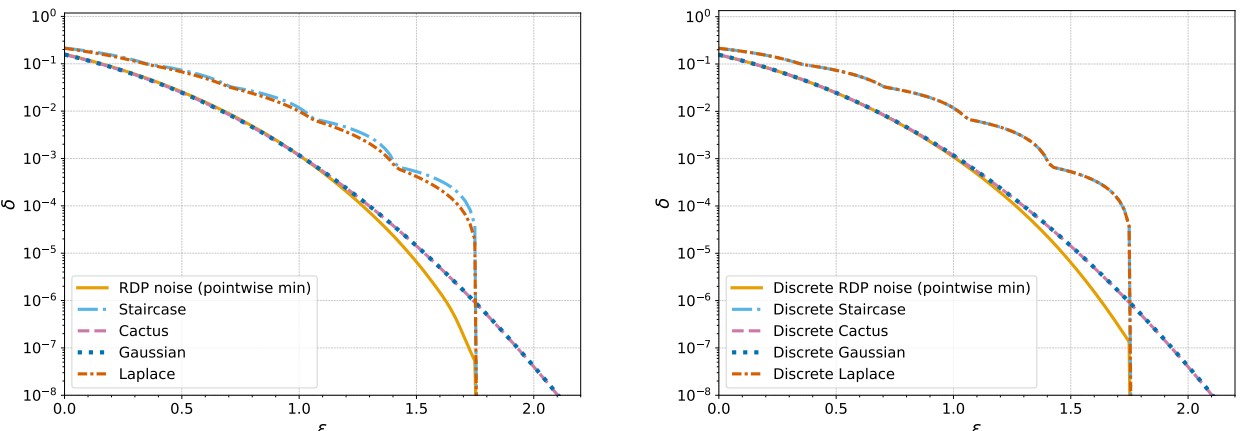

*Figure 5.* The plots compare the pointwise minimum of privacy curves from our optimized noise distributions—designed for 10 compositions, sensitivity 1, and varying $\delta$—with privacy curves for Gaussian, Laplace, Cactus, and Staircase mechanisms, all with a fixed standard deviation of $\sigma = 8$. The left plot shows continuous distributions; the right plot shows discrete distributions. All curves are evaluated at 10 compositions.

**Preconditioning:** Standard gradient descent is ill-suited to this optimization problem. The reason is that the probability vector $\mathbf{p}$ is constrained to be non-negative, in addition to the normalization and cost constraints. All these constraints by themselves can be handled by projected gradient descent (indeed, this is our approach to dealing with the normalization and cost constraints; see below). However, doing so will tend to produce $\mathbf{p}$ vectors with zeros, which lead to infinite values of the objective in (23). This blowing-up occurs because the Rényi divergence between distributions with different supports is infinite. Thus, it is not enough to keep $\mathbf{p}$ in the feasible set, it must be kept strictly feasible. This can be addressed using standard gradient descent by reducing the step size, but this causes very slow convergence. To address this limitation, we instead use *preconditioned* gradient descent, where we apply a linear transformation to the solution space, and take a gradient step in the transformed space. In particular, we use the

transformation matrix $\mathbf{M} = \text{diag}(\mathbf{p}_{k-1})^{-1}$, where $\mathbf{p}_{k-1}$ is the current optimization variable. This transformation introduces uniformity, ensuring that $\mathbf{p}$ maintains a roughly consistent distance from the boundary imposed by the non-negativity constraint.

**Gradient Calculation and Projection:** After computing the gradient in the transformed basis, we project it onto the region that meets the cost and normalization constraints. Satisfying the equation $\mathbf{A}\mathbf{p} = \mathbf{b}$ for a vector $\mathbf{p}$ in the original basis is equivalent to satisfying $\mathbf{A}\mathbf{M}^{-1}\mathbf{q} = \mathbf{b}$ for a vector $\mathbf{q}$ in the transformed basis.

**Backtracking Line Search:** The current position $\mathbf{p}_{k-1}$ in the original basis maps to $\mathbf{M}\mathbf{p}_{k-1} = \text{diag}(\mathbf{p}_{k-1})^{-1}\mathbf{p}_{k-1} = \mathbb{1}$ in the transformed basis. Thus, in each update within the transformed basis, the current position is represented by an all-ones vector. The update rule is: $\mathbf{q} = \mathbb{1} - \mu \, \mathbf{g}^{\text{proj}}$, where $\mu$ is the learning rate. The

*Table 1.* MSE improvement (%) of RDP noise over Gaussian for $\delta = 10^{-6}$ and 10 queries (mean $\pm$ std across 20 seeds).

| Dataset | $\varepsilon = 0.62$ | 0.69 | 0.78 | 0.84 | 0.97 | 1.05 |
|---|---|---|---|---|---|---|
| Breast Cancer | 8.28±0.19 | 9.14±0.19 | 9.63±0.23 | 8.61±0.17 | 10.34±0.20 | 11.48±0.18 |
| Diabetes | 8.11±0.19 | 9.06±0.21 | 9.43±0.16 | 8.48±0.17 | 10.06±0.17 | 11.12±0.22 |
| UCI Heart Disease | 8.14±0.22 | 9.05±0.16 | 9.50±0.26 | 8.52±0.19 | 10.20±0.20 | 11.31±0.18 |

corresponding update in the original basis is: $\mathbf{p}_k = \mathbf{M}^{-1}\mathbf{q}$.

The constraint $\mathbf{p}_k \geq 0$ implies $\mathbf{q} \geq 0$, leading to the upper bound $\mu^{\text{ub}} = \min_{i \in [N], \mathbf{g}_i^{\text{proj}} > 0} \frac{1}{\mathbf{g}_i^{\text{proj}}}$.

We implement a backtracking line search strategy that evaluates learning rates in the sequence $\mu^{\text{ub}}, \mu^{\text{ub}}/2, \ldots, \mu^{\text{ub}}/2^{10}$. Among these, the learning rate that achieves the largest reduction in the objective function is selected.

## 5. Numerical Results

In this section, we present numerical results evaluating the privacy characteristics of our proposed RDP noise. All $(\varepsilon, \delta)$-DP guarantees presented here are computed using the Connect-the-Dots accounting (Doroshenko et al., 2022).

Figures 2 and 3 show optimized RDP noise distributions across different settings. In particular, Figure 2 highlights how our framework automatically recovers Cactus and Staircase distributions in the regimes where they perform best ($\alpha$ close to 1 and $\infty$ respectively).

Figure 4 illustrates what we call the "moderate composition regime," showing the range of compositions and noise standard deviations ($\sigma$) where our optimized noise distribution achieves at least a 2% improvement in $\varepsilon$ compared to other known mechanisms (Gaussian, Laplace, Staircase, and Cactus) for $\delta = 10^{-6}$. The regions with more than two percent improvement depend on both $\sigma$ and the number of compositions. Importantly, there is no single mechanism that is always best across all parameter settings—so finding the best one typically requires testing all four for each combination. Even in cases where another distribution is labeled as the best, our noise still consistently performs better than the rest, although the margin is less than 2%. Our noise is beneficial not only when it gives a clear 2% or greater advantage, but also in situations where it is unclear in advance which mechanism will perform best. For example, even though the Staircase distribution is provably optimal only for a single composition and in the pure DP limit ($\delta \to 0$), it sometimes outperforms the others at $\delta = 10^{-6}$ for certain $\sigma$ and composition choices—yet it is easy to mistakenly pick a different baseline like Laplace. Our optimized noise is robust and helps avoid mistakes when choosing among existing mechanisms. Note that this plot is specific to $\delta = 10^{-6}$; for other values of $\delta$, there are additional combinations of $\sigma$ and compositions where our noise provides similar improvements,

some of which are shown in Appendix D.

In the two parts of Figure 5, we fix the number of compositions and standard deviation while optimizing the noise distribution for different $\delta$ values, for both continuous (left) and discrete (right) domains. Each $\delta$ results in a different optimized noise distribution, generating its own privacy curve. The RDP noise curves represent the pointwise minimum of these curves: that is, each point $(\varepsilon, \delta)$ on this curve is derived from the noise distribution optimized for that specific $\delta$. In contrast, Figure 1 shows the privacy curve for a specific noise distribution optimized for a single target $\delta$. Our approach remains optimal across all $\delta$ values, closely matching the performance of other noise mechanisms in their respective optimal regimes while significantly outperforming them elsewhere. Note that the range of $\varepsilon$ where we observe the greatest improvement varies with the choice of $\sigma$ and the number of compositions. We provide an additional set of plots for a different combination in Appendix D.

In Table 1, we report the performance improvement of our optimized noise over the Gaussian baseline on three widely used datasets for privacy-preserving machine learning: Breast Cancer Wisconsin (Diagnostic) (Wolberg et al., 1993), Diabetes (learn developers), and the UCI Heart Disease dataset (Cleveland subset) (Janosi et al., 1988). Details about the datasets and the experimental setup are provided in Appendix D due to space constraints. Since Gaussian noise consistently outperformed Laplace, Cactus, and Staircase mechanisms across all tested settings (with $\delta = 10^{-6}$, 10 queries, and a range of target $\varepsilon$ values), we focus our comparison on the Gaussian baseline. As shown, replacing Gaussian noise with our optimized mechanism yields an improvement of 8% to 12% across the evaluated $\varepsilon$ values.

## 6. Conclusion

We have introduced a unified optimization framework to identify optimal continuous and discrete noise distributions for $(\varepsilon, \delta)$-DP for a given cost and number of compositions. To address the problem's non-convexity, we have converted the objective into Rényi DP of an optimized order $\alpha$, yielding a finite-dimensional convex optimization problem. We have introduced a novel preconditioned gradient descent algorithm to efficiently solve this optimization problem. The resulting noise distributions are consistently better than Gaussian and Laplace distributions, with significant improvements in the moderate composition regime.

## Acknowledgements

We thank the anonymous reviewers for their valuable feedback, which significantly improved the quality of this paper. This work was supported by the National Science Foundation under Grant Nos. CIF-2312666 and CIF-2312667.

## Impact Statement

This paper presents work whose goal is to advance the field of Machine Learning. There are many potential societal consequences of our work, none which we feel must be specifically highlighted here.

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

# A. Proof of Theorem 3.1

Let $\mathcal{Z}$ be the domain of the query's output, which may consist of real numbers or integers, and let $\mathcal{P}(\mathcal{Z})$ denote the set of all probability distributions supported on $\mathcal{Z}$. Let the distribution $P_{-Z}$ represents the reflection of $P_Z$ with respect to the $y$-axis, such that $P_{-Z}(z) = P_Z(-z)$ for all $z \in \mathcal{Z}$. The feasible region $\mathcal{P}_f \subseteq \mathcal{P}(\mathcal{Z})$ encompasses all noise distributions $P_Z \in \mathcal{P}(\mathcal{Z})$ that satisfy the normalization constraint $\mathbb{E}_{P_Z}[1] = 1$, the positivity constraint $P_Z(z) \geq 0$ for all $z \in \mathcal{Z}$, and the cost constraint $\mathbb{E}_{P_Z}[c(Z)] \leq C$. The set $\mathcal{S}$ represent the possible shifts: specifically, $\mathcal{S} = [-s, s]$ if $\mathcal{Z} = \mathbb{R}$, and $\mathcal{S} = \{-s, \ldots, 0, \ldots, s\}$ if $\mathcal{Z} = \mathbb{Z}$.

Recall that

$$g_\alpha(P_Z, t) = \mathbb{E}_{P_Z}\left(\frac{P_Z}{T_t P_Z}\right)^{\alpha-1}, \tag{26}$$

and

$$g_\alpha(P_Z) = \max_{t \in \mathcal{S}} g_\alpha(P_Z, t). \tag{27}$$

To justify restricting the search to symmetric noise distributions, we demonstrate that the feasible region $\mathcal{P}_f$ is both symmetric and convex. Additionally, we prove that $g_\alpha(P_Z)$ exhibits symmetry in $Z$ and convexity in $P_Z$. By leveraging these properties, we establish that minimizing over the entire feasible region yields the same result as minimizing solely over the symmetric distributions within that region.

**Convexity of the Feasible Region:** Let $\lambda \in (0, 1)$ and $P_Z, Q_Z \in \mathcal{P}_f$, we show that $\lambda P_Z + (1-\lambda)Q_Z \in \mathcal{P}_f$. The convex combination $\lambda P_Z + (1-\lambda)Q_Z$ is a valid distribution, as it satisfies the normalization constraint as follows:

$$\mathbb{E}_{\lambda P_Z + (1-\lambda)Q_Z}[1] = \lambda\,\mathbb{E}_{P_Z}[1] + (1-\lambda)\,\mathbb{E}_{Q_Z}[1] = 1, \tag{28}$$

and it is non-negative for all $z \in \mathcal{Z}$, i.e., $\lambda P_Z(z) + (1-\lambda)Q_Z(z) \geq 0$.

We also have

$$\mathbb{E}_{\lambda P_Z + (1-\lambda)Q_Z}[c(Z)] = \lambda\,\mathbb{E}_{P_Z}[c(Z)] + (1-\lambda)\,\mathbb{E}_{Q_Z}[c(Z)] \leq \lambda C + (1-\lambda)C = C. \tag{29}$$

The inequality holds because $P_Z, Q_Z \in \mathcal{P}_f$, and thus $\mathbb{E}_{P_Z}[c(Z)] \leq C$ and $\mathbb{E}_{Q_Z}[c(Z)] \leq C$. So, we have

$$\lambda P_Z + (1-\lambda)Q_Z \in \mathcal{P}_f,$$

and the feasible region $\mathcal{P}_f$ is convex.

**Symmetry of the Feasible Region:** We show that if $P_Z$ lies within the feasible region, its reflection $P_{-Z}$ must also belong to the feasible region. If $P_Z \in \mathcal{P}(\mathcal{Z})$ (i.e., it is a non-negative, normalized measure), then $P_{-Z} \in \mathcal{P}(\mathcal{Z})$. Thus, our focus is on verifying the cost constraint. Given that $\mathbb{E}_{P_Z}[c(Z)] \leq C$, we need to show that $\mathbb{E}_{P_{-Z}}[c(Z)] \leq C$ as well. We have

$$\mathbb{E}_{P_{-Z}}[c(Z)] = \mathbb{E}_{P_Z}[c(-Z)] = \mathbb{E}_{P_Z}[c(Z)] \leq C \tag{30}$$

where the first equality follows from a variable change and the symmetry of $\mathcal{Z}$. The second equality follows from the symmetry of the cost function.

**Symmetry and Convexity of the inner maximization:**. As shown in (van Erven & Harremos, 2014) (Proof of Theorem 13), the expression $\mathbb{E}_{P_Z}\left(\frac{P_Z}{Q_Z}\right)^{\alpha-1}$ is jointly convex in $(P_Z, Q_Z)$. In (26), with $Q_Z = T_t P_Z$ and the linearity of $T_t$, it follows that $g_\alpha(P_Z, t)$ is convex in $P_Z$. Moreover, since the pointwise maximum over $t \in \mathcal{S}$ preserves convexity, $g_\alpha(P_Z)$ is also convex in $P_Z$. We now show that $g_\alpha(P_Z)$ is symmetric as follows:

$$g_\alpha(P_{-Z}) = \max_{t \in \mathcal{S}} g_\alpha(P_{-Z}, t) \tag{31}$$

$$= \max_{t \in \mathcal{S}} \mathbb{E}_{P_{-Z}}\left(\frac{P_{-Z}}{T_t P_{-Z}}\right)^{\alpha-1} \tag{32}$$

$$= \max_{t \in \mathcal{S}} \mathbb{E}_{P_Z}\left(\frac{P_Z}{T_{-t} P_Z}\right)^{\alpha-1} \tag{33}$$

$$= \max_{t \in \mathcal{S}} \; \mathbb{E}_{P_Z} \left( \frac{P_Z}{T_t P_Z} \right)^{\alpha - 1} \tag{34}$$

$$= \max_{t \in \mathcal{S}} \; g_\alpha(P_Z, t) \tag{35}$$

$$= g_\alpha(P_Z), \tag{36}$$

where (33) follows from the symmetry of $\mathcal{Z}$ and the variable substitution from $-z$ to $z$, while (34) results from the inherent symmetry of $\mathcal{S}$.

**Sufficiency of Restricting the Search to Symmetric Noise Distributions:** Leveraging these properties, we now establish that it is sufficient to restrict the search to symmetric noise distributions. Let $P_Z^*$ be an optimal noise distribution; i.e., $P_Z^*$ minimizes $g_\alpha(P_Z)$ over all $P_Z \in \mathcal{P}_f$. Also let $M^* = g_\alpha(P_Z^*)$ be the optimal objective value. Let $\lambda \in (0, 1)$, since the feasible region is convex, we have

$$\lambda P_Z^* + (1 - \lambda) P_{-Z}^* \in \mathcal{P}_f. \tag{37}$$

Since $g_\alpha(P_Z)$ is symmetric, we have $g_\alpha(P_Z^*) = g_\alpha(P_{-Z}^*) = M^*$, and so $P_{-Z}^*$ is also an optimal noise distribution. Considering the convexity of $g(P_Z)$, we have

$$g_\alpha \left( \lambda P_Z^* + (1 - \lambda) P_{-Z}^* \right) \le \lambda g_\alpha(P_Z^*) + (1 - \lambda) g_\alpha(P_{-Z}^*) = \lambda M^* + (1 - \lambda) M^* = M^* \tag{38}$$

which means every noise distribution on the line segment connecting $P_Z^*$ and $P_{-Z}^*$ is an optimal noise distribution. For the special case of $\lambda = \frac{1}{2}$, we get an optimal noise distribution

$$\frac{P_Z^* + P_{-Z}^*}{2}$$

which is symmetric. This proves the existence of a symmetric noise distribution among the set of all possible solutions and so the sufficiency of searching over only symmetric noise distributions.

## B. Proof of Proposition 3.5

For the continuous case, we have

$$\mathbb{E}[c(Z)] = \int_{z \in \mathbb{R}} c(z) \, f_Z(z) \, dz \tag{39}$$

$$= \sum_{i \in \mathbb{Z}} \int_{z \in I_i} \frac{p_{|i|}}{\Delta} c(z) \, dz \tag{40}$$

$$= \sum_{i \in \mathbb{Z}} \frac{p_{|i|}}{\Delta} \int_{z \in I_i} c(z) \, dz \tag{41}$$

$$= \frac{p_0}{\Delta} \int_{z \in I_0} c(z) \, dz + 2 \sum_{i=1}^{N-1} \frac{p_i}{\Delta} \int_{z \in I_i} c(z) \, dz + 2 \sum_{i=N}^{\infty} \frac{p_N}{\Delta} r^{i-N} \int_{z \in I_i} c(z) \, dz \tag{42}$$

$$= p_0 A_0 + 2 \sum_{i=1}^{N-1} p_i A_i + 2 p_N \sum_{i=N}^{\infty} r^{i-N} A_i, \tag{43}$$

where

$$A_i = \frac{1}{\Delta} \int_{z \in I_i} c(z) \, dz, \tag{44}$$

and (42) follows from the symmetry of $c(z)$. When $c(z) = z^2$, $A_i$ simplifies to:

$$A_i = \frac{1}{\Delta} \int_{z \in I_i} z^2 \, dz = \frac{\Delta^2}{3} \left( (i + \frac{1}{2})^3 - (i - \frac{1}{2})^3 \right) = \Delta^2 \left( i^2 + \frac{1}{12} \right), \tag{45}$$

and so

$$\text{Var}(Z) = \frac{p_0 \Delta^2}{12} + 2 \sum_{i=1}^{N-1} p_i \, \Delta^2 \left( i^2 + \frac{1}{12} \right) + 2p_N \sum_{i=N}^{\infty} r^{i-N} \Delta^2 \left( i^2 + \frac{1}{12} \right) \tag{46}$$

$$= \frac{\Delta^2}{12} \left( p_0 + \sum_{i=1}^{N-1} 2p_i + \sum_{i=N}^{\infty} 2p_N r^{i-N} \right) + 2 \sum_{i=1}^{N-1} p_i \, \Delta^2 i^2 + 2p_N \sum_{i=N}^{\infty} r^{i-N} \Delta^2 i^2 \tag{47}$$

$$= \frac{\Delta^2}{12} \left( p_0 + \sum_{i=1}^{N-1} 2p_i + \frac{2p_N}{1-r} \right) + 2 \sum_{i=1}^{N-1} p_i \, \Delta^2 i^2 + 2p_N \sum_{i=N}^{\infty} r^{i-N} \Delta^2 i^2 \tag{48}$$

$$= \frac{\Delta^2}{12} + 2\Delta^2 \sum_{i=1}^{N-1} p_i \, i^2 + 2p_N \Delta^2 \sum_{i=N}^{\infty} r^{i-N} i^2, \tag{49}$$

where (49) follows from the normalization constraint in (16), and we have

$$\sum_{i=N}^{\infty} r^{i-N} \, i^2 = \frac{r^2(N-1)^2 + N^2(1-2r) + r(2N+1)}{(1-r)^3}. \tag{50}$$

The proof for the discrete case follows a similar approach.

## C. Proof of Theorem 3.6

Here, we present a proof for the continuous case. The proof for the discrete case follows a similar approach to that of the continuous case. Recall that in the continuous case, the PDF is given by

$$f_{\mathbf{p},r,N,\Delta}(z) := \frac{p_i}{\Delta}, \quad \text{for } z \in I_i, \text{ with } i \in \mathbb{Z}, \tag{51}$$

where $I_i = \left( (i - \frac{1}{2})\Delta, (i + \frac{1}{2})\Delta \right)$, $p_{-i} = p_i$, and $p_i = p_N r^{|i|-N}$ for $|i| \geq N$. To simplify the expression inside the logarithm of the RDP, as presented in (12), we begin by substituting the piecewise-constant form of the PDF and simplifying it for any shift $t \in [-s, s]$. After substitution, the resulting expression becomes piecewise linear in $t$, with breakpoints occurring when $t$ is a multiple of the bin width $\Delta$, aligning the bins of the PDF and its shifted version. Thus, to find the maximum over $t$, we only need to consider the breakpoints within $[-s, s]$ and the interval's endpoints, forming a finite set of candidates. Finally, we simplify the expression for each element in this finite set.

For the continuous case, we have

$$g_\alpha(f_{\mathbf{p},r,N,\Delta}, t) = \int_{\mathbb{R}} f_{\mathbf{p},r,N,\Delta}(z)^\alpha \, f_{\mathbf{p},r,N,\Delta}(z-t)^{1-\alpha} dz \tag{52}$$

$$= \int_{\mathbb{R}} \left( \sum_{i=-\infty}^{\infty} \frac{p_i}{\Delta} \mathbb{1}\{z \in I_i\} \right)^\alpha \left( \sum_{j=-\infty}^{\infty} \frac{p_j}{\Delta} \mathbb{1}\{z - t \in I_j\} \right)^{1-\alpha} dz \tag{53}$$

$$= \frac{1}{\Delta} \int_{\mathbb{R}} \sum_{i=-\infty}^{\infty} p_i^\alpha \mathbb{1}\{z \in I_i\} \sum_{j=-\infty}^{\infty} p_j^{1-\alpha} \mathbb{1}\{z - t \in I_j\} \, dz \tag{54}$$

$$= \frac{1}{\Delta} \int_{\mathbb{R}} \sum_{i=-\infty}^{\infty} \sum_{j=-\infty}^{\infty} p_i^\alpha p_j^{1-\alpha} \mathbb{1}\{z \in I_i\} \mathbb{1}\{z - t \in I_j\} \, dz \tag{55}$$

$$= \frac{1}{\Delta} \sum_{i=-\infty}^{\infty} \sum_{j=-\infty}^{\infty} p_i^\alpha p_j^{1-\alpha} \int_{\mathbb{R}} \mathbb{1}\{z \in I_i\} \mathbb{1}\{z - t \in I_j\} \, dz. \tag{56}$$

Equation (56) follows from applying Tonelli's theorem for non-negative measurable functions (Wikipedia, b) twice. This holds true because both the Lebesgue measure on the real numbers and the counting measure on the integers are $\sigma$-finite, and the expression $p_i^\alpha p_j^{1-\alpha} \mathbb{1}\{z \in I_i\} \mathbb{1}\{z - t \in I_j\}$ is non-negative for all $i, j \in \mathbb{Z}$ and $z \in \mathbb{R}$. The integral within the

expression above can be simplified as follows:

$$\int_{\mathbb{R}} \mathbb{1}\{z \in I_i\}\mathbb{1}\{z - t \in I_j\}\, dz = \begin{cases} t + (j - i + 1)\Delta, & \text{if } (i - j - 1)\Delta \le t \le (i - j)\Delta, \\ -t + (i - j + 1)\Delta, & \text{if } (i - j)\Delta \le t \le (i - j + 1)\Delta, \\ 0, & \text{otherwise} \end{cases} \tag{57}$$

$$= (t + (j - i + 1)\Delta)\,\mathbb{1}\{t \in ((i - j - 1)\Delta, (i - j)\Delta]\}$$
$$+ (-t + (i - j + 1)\Delta)\,\mathbb{1}\{t \in ((i - j)\Delta, (i - j + 1)\Delta)\} \tag{58}$$

Substituting the expression from (58), the expression in (56) simplifies to:

$$\frac{1}{\Delta} \sum_{i=-\infty}^{\infty} \sum_{j=-\infty}^{\infty} p_i^\alpha \, p_j^{1-\alpha} \int_{\mathbb{R}} \mathbb{1}\{z \in I_i\}\,\mathbb{1}\{z - t \in I_j\}\, dz \tag{59}$$

$$= \frac{1}{\Delta} \sum_{j=-\infty}^{\infty} \sum_{i=-\infty}^{\infty} p_i^\alpha \, p_j^{1-\alpha} \left[ (t + (j - i + 1)\Delta)\,\mathbb{1}\{t \in ((i - j - 1)\Delta, (i - j)\Delta]\} \right.$$

$$\left. + (-t + \Delta(i - j + 1))\,\mathbb{1}\{t \in ((i - j)\Delta, (i - j + 1)\Delta)\} \right] \tag{60}$$

$$= \frac{1}{\Delta} \sum_{j=-\infty}^{\infty} \sum_{m=-\infty}^{\infty} p_{m+j}^\alpha \, p_j^{1-\alpha} \left[ (t + (-m + 1)\Delta)\,\mathbb{1}\{t \in ((m - 1)\Delta, m\Delta]\} \right.$$

$$\left. + (-t + (m + 1)\Delta)\,\mathbb{1}\{t \in (m\Delta, (m + 1)\Delta)\} \right] \tag{61}$$

where, in (61), we apply the variable change $m = i - j$ for each fixed $j$. Since $t$ is a real number in the interval $[-s, s]$, the indicator functions $\mathbb{1}\{t \in ((m - 1)\Delta, m\Delta]\}$ and $\mathbb{1}\{t \in (m\Delta, (m + 1)\Delta)\}$ are zero for values of $m$ outside the set $\{-\frac{s}{\Delta}, \ldots, 0, \ldots, \frac{s}{\Delta}\}$, where $\Delta$ is chosen such that $\frac{s}{\Delta}$ is an integer. Therefore, $g_\alpha(f_{\mathbf{p},r,N,\Delta}, t)$ simplifies to:

$$g_\alpha(f_{\mathbf{p},r,N,\Delta}, t) = \frac{1}{\Delta} \sum_{j=-\infty}^{\infty} \sum_{m=-\frac{s}{\Delta}}^{\frac{s}{\Delta}} p_{m+j}^\alpha \, p_j^{1-\alpha} \left[ (t + (-m + 1)\Delta)\,\mathbb{1}\{t \in ((m - 1)\Delta, m\Delta]\} \right.$$

$$\left. + (-t + (m + 1)\Delta)\,\mathbb{1}\{t \in (m\Delta, (m + 1)\Delta)\} \right]. \tag{62}$$

To solve the maximization problem over $t \in [-s, s]$, observe that the above expression is piecewise linear in $t$. Therefore, it is sufficient to evaluate the maximum at the endpoints of the linear segments. These endpoints are given by $t \in \left\{ a\Delta \mid a \in \left\{ -\frac{s}{\Delta}, \ldots, 0, \ldots, \frac{s}{\Delta} \right\} \right\}$. Let evaluate the expression in (62) at $t = a\Delta$:

$$g_\alpha(f_{\mathbf{p},r,N,\Delta}, a\Delta)$$

$$= \frac{1}{\Delta} \sum_{j=-\infty}^{\infty} \sum_{m=-\frac{s}{\Delta}}^{\frac{s}{\Delta}} p_{m+j}^\alpha \, p_j^{1-\alpha} \left[ (a\Delta + (-m + 1)\Delta)\,\mathbb{1}\{a\Delta \in ((m - 1)\Delta, m\Delta]\} \right.$$

$$\left. + (-a\Delta + (m + 1)\Delta)\,\mathbb{1}\{a\Delta \in (m\Delta, (m + 1)\Delta)\} \right] \tag{63}$$

$$= \frac{1}{\Delta} \sum_{j=-\infty}^{\infty} p_{a+j}^\alpha \, p_j^{1-\alpha} (a\Delta + (-a + 1)\Delta) \tag{64}$$

$$= \sum_{j=-\infty}^{\infty} p_{a+j}^\alpha \, p_j^{1-\alpha} \tag{65}$$

$$= \sum_{j=-\infty}^{\min\{-N-a,-N\}-1} (p_N \, r^{-a-j-N})^\alpha \, (p_N \, r^{-j-N})^{1-\alpha} + \sum_{j=\min\{-N-a,-N\}}^{\max\{N,N-a\}} p_{a+j}^\alpha \, p_j^{1-\alpha} +$$

$$+ \sum_{j=\max\{N,N-a\}+1}^{\infty} (p_N \, r^{a+j-N})^\alpha \, (p_N \, r^{j-N})^{1-\alpha} \tag{66}$$

$$= p_N \, r^{-a\alpha-N} \sum_{j=-\infty}^{\min\{-N-a,-N\}-1} r^{-j} + \sum_{j=\min\{-N-a,-N\}}^{\max\{N,N-a\}} p_{a+j}^\alpha \, p_j^{1-\alpha} + p_N \, r^{\alpha a - N} \sum_{j=\max\{N,N-a\}+1}^{\infty} r^j \tag{67}$$

$$= p_N \, r^{-a\alpha-N} \, \frac{r^{-\min\{-N-a,-N\}+1}}{1-r} + \sum_{j=\min\{-N-a,-N\}}^{\max\{N,N-a\}} p_{a+j}^\alpha \, p_j^{1-\alpha} + p_N \, r^{\alpha a - N} \, \frac{r^{\max\{N,N-a\}+1}}{1-r} \tag{68}$$

$$= \frac{p_N \, r}{1-r} \left( r^{-a\alpha-N+\max\{N+a,N\}} + r^{a\alpha-N+\max\{N,N-a\}} \right) + \sum_{j=\min\{-N-a,-N\}}^{\max\{N,N-a\}} p_{a+j}^\alpha \, p_j^{1-\alpha} \tag{69}$$

$$= \frac{p_N \, r}{1-r} \left( r^{-a\alpha+\max\{a,0\}} + r^{a\alpha+\max\{0,-a\}} \right) + \sum_{j=\min\{-N-a,-N\}}^{\max\{N,N-a\}} p_{a+j}^\alpha \, p_j^{1-\alpha} \tag{70}$$

$$= \frac{p_N \, r}{1-r} \left( r^{(1-\alpha)|a|} + r^{\alpha|a|} \right) + \sum_{j=-N}^{N-|a|} p_{|a|+j}^\alpha \, p_j^{1-\alpha} + p_N^\alpha r^{\alpha(|a|-N)}$$

$$\sum_{j=N-|a|+1}^{N} r^{\alpha j} \, p_j^{1-\alpha} + \left( p_N^{1-\alpha} r^{-N(1-\alpha)} \sum_{j=-|a|-N}^{-N-1} p_{|a|+j}^\alpha \, r^{-j(1-\alpha)} \right) \mathbb{1}\{a \neq 0\} \tag{71}$$

where (64) follows because an open interval $(m\Delta, (m+1)\Delta)$ does not include any endpoints. So

$$\mathbb{1}\{a\Delta \in (m\Delta, (m+1)\Delta)\} = 0$$

and a half-open interval $((m-1)\Delta, m\Delta]$ includes the endpoint $a\Delta$ if $m = a$. The equality (71) follows because

$$r^{-a\alpha+\max\{a,0\}} + r^{a\alpha+\max\{0,-a\}} = \begin{cases} r^{-a\alpha+a} + r^{a\alpha} & \text{if } a \geq 0 \\ r^{-a\alpha} + r^{a\alpha-a} & \text{if } a < 0 \end{cases} = r^{(1-\alpha)|a|} + r^{\alpha|a|} \tag{72}$$

and

$$\sum_{j=\min\{-N-a,-N\}}^{\max\{N,N-a\}} p_{a+j}^\alpha \, p_j^{1-\alpha} \tag{73}$$

$$= \sum_{j=-N}^{N} p_{|a|+j}^\alpha \, p_j^{1-\alpha} + \left[ p_N^{1-\alpha} r^{-N(1-\alpha)} \sum_{j=-|a|-N}^{-N-1} p_{|a|+j}^\alpha \, r^{-j(1-\alpha)} \right] \mathbb{1}\{a \neq 0\} \tag{74}$$

$$= \sum_{j=-N}^{N-|a|} p_{|a|+j}^\alpha \, p_j^{1-\alpha} + p_N^\alpha r^{\alpha(|a|-N)} \sum_{j=N-|a|+1}^{N} r^{\alpha j} \, p_j^{1-\alpha}$$

$$+ \left[ p_N^{1-\alpha} r^{-N(1-\alpha)} \sum_{j=-|a|-N}^{-N-1} p_{|a|+j}^\alpha \, r^{-j(1-\alpha)} \right] \mathbb{1}\{a \neq 0\}. \tag{75}$$

where we have used the assumption that the distribution is symmetric, meaning $p_i = p_{-i}$.
The quantity in (71) is symmetric with respect to $a$; therefore, the maximum over

$$t \in \left\{ a\Delta \mid a \in \left\{ -\frac{s}{\Delta}, \dots, 0, \dots, \frac{s}{\Delta} \right\} \right\}$$

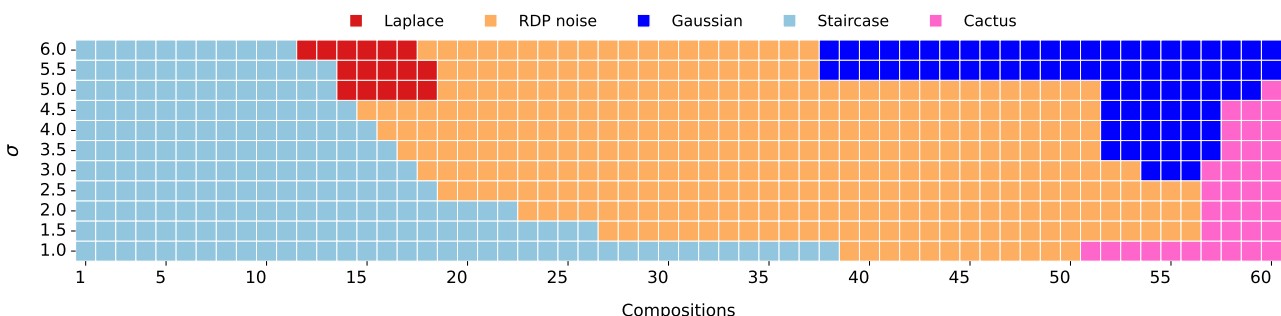

Figure 6. For fixed $\delta = 10^{-9}$, the grid evaluates different combinations of composition count and noise standard deviation ($\sigma$), marking RDP noise as the winner whenever it achieves more than a 2% improvement in $\varepsilon$ over the best alternative (among Gaussian, Laplace, Staircase, and Cactus). Even when another distribution is marked as the best, our noise still consistently outperforms the others, although by less than 2%.

reduces to a maximum over

$$t \in \left\{ a\Delta \mid a \in \left\{ 0, \ldots, \frac{s}{\Delta} \right\} \right\},$$

and $|a|$ simplifies to $a$. Moreover, when $a = 0$, the two distributions are identical, and the divergence reaches its minimum value of zero. Hence, we can exclude $a = 0$ from the set. So, the task of finding the worst-case shift collapses to the following optimization:

$$\max_{a \in \left\{ 1, \ldots, \frac{s}{\Delta} \right\}} \frac{p_N \, r}{1 - r} \left( r^{(1-\alpha)a} + r^{\alpha a} \right) + \sum_{j=-N}^{N-a} p_{a+j}^{\alpha} \, p_j^{1-\alpha} + p_N^{\alpha} r^{\alpha(a-N)}$$

$$\sum_{j=N-a+1}^{N} r^{\alpha j} \, p_j^{1-\alpha} + p_N^{1-\alpha} r^{-N(1-\alpha)} \sum_{j=-a-N}^{-N-1} p_{a+j}^{\alpha} \, r^{-j(1-\alpha)}. \tag{76}$$

Utilizing the symmetry of the noise distribution ($p_i = p_{-i}$), we can rewrite the maximization as follows:

$$\max_{a \in \left\{ 1, \ldots, \frac{s}{\Delta} \right\}} \frac{p_N \, r}{1 - r} \left( r^{(1-\alpha)a} + r^{\alpha a} \right) + \sum_{j=-N}^{N-a} p_{|a+j|}^{\alpha} \, p_{|j|}^{1-\alpha} + p_N^{\alpha} r^{\alpha(a-N)}$$

$$\sum_{j=N-a+1}^{N} r^{\alpha j} \, p_{|j|}^{1-\alpha} + p_N^{1-\alpha} r^{-N(1-\alpha)} \sum_{j=-a-N}^{-N-1} p_{|a+j|}^{\alpha} \, r^{-j(1-\alpha)}. \tag{77}$$

## D. Additional Numerical Results

The region of compositions and noise levels ($\sigma$) where our optimized noise achieves at least a 2% improvement in $\varepsilon$ over standard mechanisms (Gaussian, Laplace, Staircase, and Cactus) shifts as $\delta$ changes. Figure 6 presents a plot similar to Figure 4 but for $\delta = 10^{-9}$. These results further highlight that the optimal noise distribution is highly setting-dependent. Since our RDP noise consistently performs well across different settings and avoids the need for manual selection among existing mechanisms, it offers a more reliable default.

Figure 7 shows a plot similar to Figure 5, but for a smaller $\sigma = 5$, with the number of compositions fixed at 10 in both cases. As shown, the range of $\varepsilon$ where our method provides the greatest improvement shifts with the choice of $\sigma$. For $\sigma = 5$ (Figure 7), the peak improvement occurs between $\varepsilon = 2$ and 2.8, while for the larger $\sigma = 8$ in Figure 5, it lies between approximately 1.2 and 1.8. This shift is expected—optimizing for a larger $\sigma$ corresponds to a higher privacy regime, resulting in smaller $\varepsilon$ values.

To obtain the results in Table 1, we assume that the 5th and 95th percentiles of each feature are privately released. These quantiles are used to rescale each feature by subtracting the 5th percentile and dividing by the difference between the 95th and 5th percentiles. This transformation maps most feature values to the $[0, 1]$ range, and any values outside this range are

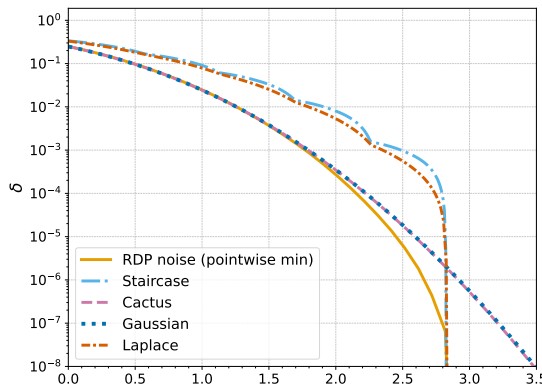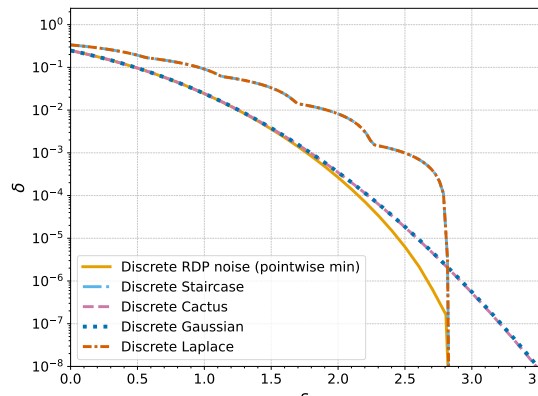

*Figure 7.* The plots compare the pointwise minimum of privacy curves from our optimized noise distributions—designed for 10 compositions, sensitivity 1, and varying $\delta$—with privacy curves for Gaussian, Laplace, Cactus, and Staircase mechanisms, all with a fixed standard deviation of $\sigma = 5$. The left plot shows continuous distributions; the right plot shows discrete distributions. All curves are evaluated at 10 compositions.

clipped to 0 or 1. This normalization step ensures that all features are on a consistent scale before noise is added. Mapping back to the original feature scale can be done using the privately released quantiles.

Since the quantile release step incurs the same privacy cost for both the Gaussian and RDP noise mechanisms, we do not account for it separately in our analysis. Instead, we focus on evaluating the privacy-utility trade-offs introduced by the noise-adding mechanisms themselves.

We then add noise using either the Gaussian mechanism or our optimized RDP noise. For each query, we generate 100,000 differentially private outputs by adding noise according to the selected mechanism and compute the mean squared error (MSE) with respect to the true (non-private) value. We average the MSEs across 10 queries, repeat the process for 20 random seeds, and report the mean and standard deviation ($\pm$) of the improvement over Gaussian noise.

For the remainder of this section, we provide details about the datasets and the queries used in the experiments presented in Table 1.

**Breast Cancer Wisconsin (Diagnostic) (Wolberg et al., 1993)**: This dataset contains 569 records with 30 continuous features extracted from digitized images of fine needle aspirates of breast masses. These features capture various properties of cell nuclei, including radius, texture, perimeter, area, smoothness, and symmetry. We define 10 continuous queries, each computing the average of a key diagnostic indicator. These include the overall average values of features such as mean radius, texture, area, smoothness, and fractal dimension. Such queries are commonly used in biomedical research and medical data analysis to support disease characterization and model interpretability in diagnostic applications.

**Diabetes dataset (learn developers)**: This dataset includes 442 patient records and 10 features representing physiological variables such as age, sex, body mass index (BMI), blood pressure, and six blood serum measurements (s1 to s6). We define 10 continuous queries, each computing the mean of a feature across the dataset. These queries capture the average body mass index, blood pressure, age, sex indicator (reflecting gender distribution), and the average values of each of the six serum biomarkers.

**UCI Heart Disease Dataset (Cleveland subset)(Janosi et al., 1988):** A widely used clinical dataset containing diagnostic information for 303 patients undergoing evaluation for coronary artery disease. It includes a mix of demographic and clinical features such as age, sex, resting blood pressure, serum cholesterol, maximum heart rate achieved during exercise, and indicators of exercise-induced angina. To simulate realistic medical analytics, we define 10 queries focused on patient demographics and cardiac health metrics. These include the average age and maximum heart rate within subgroups defined by sex and angina status, as well as the average resting blood pressure and cholesterol levels for higher-risk populations, such as individuals with hypertension or adults over the age of 55.

