# OpenReview forum: "Optimizing Noise Distributions for Differential Privacy"
_ICML.cc/2025/Conference — ICML 2025 poster_

### Official Review · Reviewer_zWjf · 2025-02-24

**Overall Recommendation:** 2

**Summary:**

The paper studies non-canonical (i.e., not Laplace or Gaussian) noise distributions for answering $d$ queries under $(\varepsilon, \delta)$-DP. It casts the overall problem as follows: the user provides $\delta$, $d$, and the sensitivity and error constraint $\sigma$ for each query. Then the provided algorithm formulates this as a convex optimization problem over discrete (or "morally" discrete) Renyi DP noise distributions that can be represented as finite real vectors and applies a carefully chosen optimization algorithm to solve for a distribution achieving the minimal possible $\varepsilon$. Depending on the provided user inputs, the resulting algorithm can produce distributions that achieve noticeable improvements over Laplace and Gaussian noise.

## update after rebuttal
I've increased my score to weak reject. The author response resolved my questions about parameters, numerical stability, and the relationship to the staircase mechanism (and I'd suggest to the authors to add these discussions to the next version of the paper). I don't think it's clearly wrong for the paper to be accepted. However, to me the small degree of improvement, the opacity of the eventual distributions, and the need to use connect-the-dot accounting for each new problem instance limit how useful the results are theoretically or practically.

**Claims And Evidence:**

The abstract claims "significant[ly]" better privacy guarantees than the Laplace and Gaussian distributions in some parameter settings. "Significant" is subjective, but the improvements appear to range from ~0 to 9%, with the latter occurring in a somewhat narrow portion of the parameter space, with some further questions about baselines (see boxes below). I think the significance is questionable.

**Essential References Not Discussed:**

N/A

**Experimental Designs Or Analyses:**

The paper omits a few things that I think should be discussed.

1) How is $N$, the support size of the distribution, chosen? What is its value in the experiments? What about $r$, $K$, and $T$?

2) How easy/fast are these algorithms to run relative to their simple Laplace and Gaussian counterparts? Runtime doesn't appear to be discussed anywhere (or I missed it).

3) If I remember correctly, the staircase mechanism dominates the Laplace mechanism, particularly for large $\varepsilon$. The paper claims that its optimization recovers the staircase mechanism in the single composition regime but otherwise omits it. It seems like the staircase mechanism should replace the Laplace mechanism in the experiments, especially since the figures show the largest improvements at large $\varepsilon$, where the staircase mechanism should also improve over the Laplace mechanism. Concretely: is the returned distribution at these settings just the staircase mechanism?

4) Can you say a bit more about possible numerical issues? The optimization problem described in Theorem 3.6 looks like it may have under/overflow issues. This is especially relevant because the DP guarantee of the algorithm is entirely dependent on the optimization working correctly.

**Methods And Evaluation Criteria:**

The form of the experiments provided, mostly plots involving some combination of $d$, $\varepsilon$, $\delta$, and $\sigma$ seems reasonable.

**Other Comments Or Suggestions:**

(See other responses.)

**Other Strengths And Weaknesses:**

(See other responses.)

**Questions For Authors:**

In addition to questions 1)-4) written above, I'll ask:

5) What do the authors think this algorithm "tells us" about DP? It returns what the authors claim is a new and better kind of distribution, but I'm curious what form the distribution takes. In addition to the question above about its relationship to the staircase mechanism, it would be interesting to see approximately what shape the distribution has, or other properties that might explain its improvement over baselines. Without that, it's hard to say how just knowing that the distributions exist improves our understanding of DP.

**Relation To Broader Scientific Literature:**

The general idea of solving an optimization problem to find a private additive noise distribution is (as the paper notes) not new. To the best of my knowledge, this formulation in terms of ~finite discrete distributions has not been studied before. I think the paper is at least moderately novel.

**Theoretical Claims:**

I didn't check any of the proofs. The basic ideas seem reasonable.

---

> ### Author Rebuttal · Authors · 2025-03-31
>
> We thank the reviewer for their constructive feedback. Below are our responses to their concerns.
>
> **The abstract claims "significant[ly]":** Since "significant" is subjective, we will clearly specify the gains in the abstract if accepted. We addressed our framework's practicality in our response to reviewer WDdd and kindly refer the reviewer to that discussion (see our responses to all their questions).
>
> **Q1)** In our framework, $N$ is the number of bins before the geometric tail begins, resulting in a probability vector of length $N+1$. $\Delta$ is the bin width, so the geometric tails start beyond $-N\Delta$ and $+N\Delta$. Figure 4 provides the values of $\Delta, r$, and $N$ for a specific distribution. Our rule of thumb for selecting these parameters is to choose a small $\Delta$ (around 0.05) and set $N$ such that $N \Delta$ is about 20 times the noise’s standard deviation to capture its behavior effectively. For $r$, we select a value close to 1 (e.g., 0.9999) to ensure the noise has a heavy tail, as a light tail may fail to satisfy the pure DP constraint. In Algorithm 3, $T$ refers to the time step for updating $\alpha$. Specifically, every $T$ iterations, $\alpha$ is updated to optimize the moments accountant. Based on our observations, a $T$ between 10 and 20 is sufficient to determine $\alpha$. We set the total number of iterations, $K$, to 5,000, but with preconditioning, convergence typically stabilizes around 2,000 iterations based on our observations. We will include these details in the final version.
>
> **Q2)** Our method is computationally efficient due to the convexity of the optimization and the use of preconditioned gradient descent, achieving an optimal distribution in 18.7 s ± 431 ms (mean ± std). While our method requires additional computation during the optimization phase compared to Laplace or Gaussian, this is a one-time cost. Once completed, sampling is fast and straightforward due to its discrete nature using inverse CDF. Specifically, for sampling 50,000 times, our noise takes 4.61 ms ± 94.2 µs, as compared to 3.37 ms ± 980 µs for Gaussian (numpy.random.normal) and 3.2 ms ± 877 µs for Laplace (numpy.random.laplace). All runtime experiments were conducted on Google Colab's CPU environment without GPU acceleration. If accepted, we will include more detailed runtime comparisons.
>
> **Q3)** The Staircase mechanism is optimal in the single composition setting for pure DP ($\delta = 0$). Since we did not consider the $\delta = 0$ case in our experiments, we compared our noise to Laplace instead. If accepted, we will compute the composed $(\epsilon, \delta)$ values for Staircase, compare them with our noise, and add figures showing how our distributions recover this mechanism. We have addressed why our framework can recover the Staircase in our response to reviewer TUrb (Q2 and Q3) and kindly ask the reviewer to refer to that rebuttal for more details.
>
> **Q4)** Since Gaussian is a straightforward baseline, our algorithm starts from a Gaussian approximation to ensure the initial point is feasible. The main challenge is preventing noise parameters $(p_0, \dots, p_N)$ from approaching zero, as Renyi DP requires full support to avoid instability. This leads to slow convergence since it requires careful steps to stay within the positive orthant. To address this, we introduced a **novel preconditioning approach** in Section 4 (end of page 7, start of page 8) to stabilize the objective, prevent numerical issues, and accelerate convergence. As a result, we do not encounter overflow or underflow issues. In particular, the iterates on the optimization variable stay feasible while continually improving the objective, after starting from a point close to the Gaussian as mentioned above. This ensures that, even if the algorithm does not fully converge (which empirically it almost always does), it will achieve a good solution. We hope this addresses the reviewer's concern. If not, we would appreciate further clarification.
>
> **Q5)** We appreciate this query. Our distribution is setting-dependent and adapts to approach the optimal shape in each regime. The existence of fundamentally different optimal distributions, such as the monotone Staircase and non-monotone Cactus, highlights the difficulty of assessing a noise distribution’s utility-privacy tradeoff based solely on its appearance. Supporting both types demonstrates our framework’s generality and optimality. The shape of our optimal distribution combines the characteristics of different distributions depending on the setting. Figure 2 implicitly illustrates how the shape of our optimal distribution transitions between Laplace and Gaussian as cost or composition increases; specifically, our distribution resembles Laplace when Laplace is close to optimal; similarly for Gaussian. We already included an example of an optimized distribution in Figure 4. We will add more illustrations of optimal distributions in the final version.

---

> > ### Comment · Reviewer_zWjf · 2025-04-01
> >
> > Q1) Thanks for clarifying the parameter ranges.
> >
> > Q2) Thanks for the timing results.
> >
> > Q3) I understand that the optimization given by this paper can recover the Staircase mechanism (or a close approximation to it). However, to my reading, the current version of the paper is unclear about whether it is recovering the Staircase mechanism in the multiple composition setting. That matters because recovering a known noise distribution is less novel than identifying a new one. The suggested experiment computing the $(\varepsilon, \delta)$-DP values and adding it as a comparison would clarify this point -- if the recovered distribution is clearly better than Staircase, that would be pretty good evidence -- but without it, I think the paper is missing a necessary baseline.
> >
> > Q4) I follow how a step like preconditioning is necessary to ensure a Renyi DP guarantee. However, I don't see how it mitigates issues like: based on the answer to Q1, $r \approx 1$ and $N \approx 400\sigma$. AFAIK, the typical range of $\alpha$ used in RDP conversions is $[1, 10]$. The condition in Equation 23 of the optimization problem includes terms like $r^{\alpha N}$. Maybe having an exponent in the thousands in this condition is fine for some reason, but it's not obvious to me at first look. As mentioned in the initial review, since the DP guarantee hinges on the optimization, this seems worth being careful about. Alternatively, if the DP guarantee of the obtained distribution can be verified in a more obviously stable way after the fact, that would mitigate this concern.
> >
> > Q5) I appreciate that the flexibility of this method is a point in its favor. But I think a possible weakness is that, since the distributions are obtained from a fairly opaque optimization process, they don't tell us much about where the improvement "comes from".

---

> > > ### Author Response · Authors · 2025-04-01
> > >
> > > We appreciate the reviewer's feedback. Here is our response to their concern.
> > >
> > > **Q3)** We appreciate the reviewer’s comment. Our observation of the optimal noise distributions in the multiple composition regime indicates that these distributions do not resemble the Staircase mechanism in this setting (see, for example, Figure 4)—we are indeed discovering new distributions. In the discrete setting with sensitivity 1, the Laplace and Staircase mechanisms are identical, resulting in the same privacy curve. The right panel of Figure 3 compares discrete mechanisms for sensitivity 1 (Laplace = Staircase, Gaussian, and our noise) over 10 compositions. In this figure, since our mechanism clearly outperforms Laplace, it follows that our derived noise is fundamentally different from the Staircase in this setting.
> > >
> > > **For the continuous case, we direct the reviewer’s attention to the plot available at this link: https://drive.google.com/file/d/1-Tv_fsx-82FQZgptxa23BqwPJAcKMUo6/view?usp=sharing. This plot compares the privacy curves of the Laplace and Staircase mechanisms under the same setting as the left panel of Figure 3 in our paper (10 compositions, standard deviation of 5, sensitivity of 1). This plot clearly illustrates why we chose to benchmark against the Laplace mechanism rather than the Staircase mechanism in the multiple composition regime. As shown, Laplace outperforms Staircase in this setting. Since our noise distribution outperforms Laplace, this confirms that our noise is not only different from but also an improvement over the Staircase mechanism. We would also like to highlight that computing the privacy guarantees under composition for the Staircase mechanism cannot be done in closed form and must be done numerically. As far as we can tell, we are the first to consider this.**
> > >
> > > As mentioned earlier, we will include a direct comparison with the Staircase mechanism across a wider range of parameters in the final version of the paper.
> > >
> > > **Q4)** As mentioned, $r$ is chosen to be very close to 1, ensuring that raising it to large powers remains manageable. Additionally, we initialize our optimization with an approximation of a Gaussian, which serves as a feasible starting point. As explained in the "Optimization for $\alpha$" part on page 7, we also initialize our $\alpha$ with the optimal $\alpha$ for the Gaussian. From this starting point, our optimization continually improves the objective and avoids moving in a direction that would result in an infinite objective value.
> > >
> > > Moreover, we emphasize that the DP guarantees presented in our paper do not stem from our algorithm itself, but from the state-of-the-art privacy accountant, Connect-the-Dots. We derive an optimal noise distribution from our algorithm and compute its privacy guarantees using this established method, which further confirms that the optimization algorithm is functioning correctly.
> > >
> > > **Q5)** We would offer the following analogy to illustrate the value of our approach: neural networks have to be trained by solving an optimization problem. Just like our problem, there is simply no way to derive the parameters in an entirely theoretical manner. This is not to say that neural networks take the theory, or human input, out of the picture: the loss function, the architecture of the network, the optimization algorithm, etc. are all crucial pieces that can be understood theoretically and contribute to the impact of neural networks, even though the optimization itself is somewhat opaque. In this analogy, the loss function, architecture, optimization choices correspond to our objective function being derived from the Moments Accountant bound using Renyi DP, the way we parameterize the distribution (piecewise constant with infinite geometric tails that maintain DP without contributing much to it), and the preconditioned gradient descent algorithm. All of these choices draw from a theoretical understanding of the setting, even though the optimization itself is somewhat opaque.

---

### Official Review · Reviewer_AB3E · 2025-03-06

**Overall Recommendation:** 2

**Summary:**

This paper addresses the optimization of noisy distributions under the RDP framework. Compared to classic approaches, such as Laplace or Gaussian mechanisms, the derived distribution achieves a lower overall cost.

**Claims And Evidence:**

Yes.

**Essential References Not Discussed:**

No.

**Experimental Designs Or Analyses:**

Yes.

**Methods And Evaluation Criteria:**

Yes.

**Other Comments Or Suggestions:**

See the question below.

**Other Strengths And Weaknesses:**

Pros: The problem is important, and the proposed approach is compelling.
Cons: Please refer to my questions for further details.

**Questions For Authors:**

1. From the numerical results presented in the figures, it appears that there is little difference for smaller values of $\epsilon$ (e.g., $\epsilon < 2$) compared to the Gaussian distribution. Meanwhile, adding Gaussian noise may be more straightforward for statistical inference or uncertainty quantification (for example, when constructing confidence intervals). Could the authors provide further justification for adopting the proposed distribution instead?

2. It might be more illustrative if the authors included an example to demonstrate the advantages of their method. For instance, it would be helpful to see a specific problem where the variance of the new privacy-preserving estimator is clearly lower than that of a traditional estimator.

**Relation To Broader Scientific Literature:**

See the pros below.

**Theoretical Claims:**

Yes.

---

> ### Author Rebuttal · Authors · 2025-04-01
>
> We appreciate the reviewer's thoughtful feedback. Below, we provide detailed responses to their concerns.
>
> **From the numerical results presented in the figures, it appears that there is little difference for smaller values of ϵ (e.g., ϵ<2) compared to the Gaussian distribution. Meanwhile, adding Gaussian noise may be more straightforward for statistical inference or uncertainty quantification (for example, when constructing confidence intervals). Could the authors provide further justification for adopting the proposed distribution instead?**
>
> We appreciate the reviewer’s careful consideration of our work. If the reviewer is referring to Figure 3 in our paper, where there is a small difference for $\epsilon<2$ , we agree with the reviewer regarding this specific plot. However, we would like to highlight that the results in this plot are sensitive to the cost threshold ($\sigma^2$) and the number of compositions. In certain settings, our noise mechanism can achieve an improvement over Gaussian noise even for $\epsilon$ values less than 2. For example, if we compare our optimized noise, designed for 10 compositions and $\delta = 10^{-6}$, against Gaussian and Laplace distributions, all with the same standard deviation of 8, at the target $\delta = 10^{-6}$, our noise achieves an $\epsilon$ of 1.62, compared to 1.76 for Laplace and 1.74 for Gaussian noise. Replacing the Gaussian or Laplace distributions with our optimized noise yields improvements of 6.89\% and 7.95\%, respectively, in the $\epsilon$ value. So, in general, it is not true that Gaussian noise should always be preferred for $\epsilon$ values less than 2.
>
> From the perspective of sampling complexity, we note that classical inverse CDF sampling methods can be applied straightforwardly to our optimal noise distribution. Thus, while our optimal noise distribution provides a better privacy-utility tradeoff than Gaussian noise in appropriate settings, its practical use remains efficient. We have included sampling time comparisons between our noise and Gaussian in the rebuttal for reviewer zWjf (response to Q2) and kindly encourage the reviewer to refer to that for further details.
>
> We have provided additional justification for why our noise should be adopted in the rebuttal for Reviewer WDdd, and we kindly ask the reviewer to refer to that section for more details (please see our responses to all questions from that reviewer).
>
> **It might be more illustrative if the authors included an example to demonstrate the advantages of their method. For instance, it would be helpful to see a specific problem where the variance of the new privacy-preserving estimator is clearly lower than that of a traditional estimator.**
>
> We thank the reviewer for this insightful comment. In our rebuttal to reviewer WDdd, we have provided details on the practicality of our framework, including applicable datasets (response to Q2) and an example illustrating how our method reduces estimator variance (response to Q3). We kindly ask the reviewer to refer to that rebuttal for further clarification.

---

> > ### Comment · Reviewer_AB3E · 2025-04-07
> >
> > I appreciate the authors' response and their efforts to improve this manuscript. After carefully reading the response to me and other reviewers, I think this manuscript indeed provides some novel results about privacy mechanisms, although it may lack some intuitive or direct motivation/application that can replace the traditional noise, such as Gaussian noise. It is a difficult decision, and for now, I have to keep the scores.

---

> > > ### Author Response · Authors · 2025-04-07
> > >
> > > We thank the reviewer for recognizing the novelty of our work. We appreciate the thoughtful feedback and the opportunity to clarify and elaborate on the contributions and significance of our work.
> > >
> > > We would like to reiterate that while Gaussian noise is widely used in practice, its popularity stems not from its universal optimality, but from its analytical convenience and tractability. However, Gaussian noise is not tailored to specific utility goals or problem constraints. In contrast, our framework provides noise distributions optimized for the specific setting at hand, leading to better privacy-utility tradeoffs.
> > >
> > > As demonstrated in the example provided in our response to Reviewer WDdd, our optimized noise achieves approximately 10\% improvement in mean squared error compared to the best of Gaussian and Laplace for the same privacy guarantee in certain settings. Furthermore, as shown in Figure 1 of our paper, our method yields about a 5\% improvement in $\varepsilon$ for the same quadratic cost—a considerable gain when working with tight privacy budgets.  These results highlight that, in the right settings, our optimized noise offers tangible benefits and is a compelling alternative to standard mechanisms.

---

### Official Review · Reviewer_TUrb · 2025-03-12

**Overall Recommendation:** 3

**Summary:**

The authors of the paper introduce an optimization framework that optimizes noise distribution for $\alpha$-RDP, where the optimal distribution can be obtained by a finite-dimensional convex optimization problem. Their main contribution is the proposal of optimized distribution for a moderate composition regime (single/large cases are already shown in previous works).

For their objective, they first formulate an optimization problem for **minimizing $\alpha$-Renyi Divergence for a given constraint (e.g., variance of the distribution)**.
Then, they show that the problem is convex and symmetric.

**Claims And Evidence:**

Throughout the paper, the authors introduce their contributions very clearly. Also, the experimental results clearly show the superiority compared to Laplace and Gaussian mechanisms.

**Essential References Not Discussed:**

None

**Experimental Designs Or Analyses:**

The reviewer has a question about the experimental design, especially for baselines. Please refer to the “Question for Authors” Section.

**Methods And Evaluation Criteria:**

Since the aim is to minimize the value of $\epsilon$ for a given value of $\delta$ and variance bound $\sigma$, the evaluation criteria is proper.

**Other Comments Or Suggestions:**

Here are my minor concerns; please refer to the next section for more detailed reviews.
- In the preliminaries (line 120-130), the definition of $\sim$ is duplicated ($X\sim P$ for probability distribution and $d\sim d’$ for the neighboring sets).
- After Eq. (11), the authors need to explain the constraint with an example (if $c(x)=x^2$ the constraint limits the variance of additive noise).

**Other Strengths And Weaknesses:**

Throughout the paper, the authors clearly introduce the novelty of their work, as well as their mathematical contributions. The preliminaries part is well-written. The idea of min-max optimization is interesting, and the approach provided in this manuscript is novel.

**Questions For Authors:**

- In the contribution parts, the authors mentioned that “the algorithm recovers as special cases noise distributions that are known to be optimal in different regimes, such as the Stair case and Cactus mechanisms.” However, the reviewer cannot find the related experimental results.
- In the numerical results, the authors said “a small number of compositions, Laplace is close to the optimal, and the Gaussian is close to the optimal if large number of compositions,” although the optimal one would be star-case and cactus distributions. Can the authors provide experimental results by adding two baselines (stair-case and cactus).
- Are there any additional practical examples regarding moderate composition regimes other than U.S. Census Bureau?
- Please better introduce the relation between the authors’ work and the machine learning. For example, how can we use moderate competition in **Machine Learning** field?

**Relation To Broader Scientific Literature:**

Previously, noise distribution optimizations for differential privacy have been discussed for a single composition or a large number of composition regimes, where the optimal strategies are stair-case and cactus mechanisms, respectively. The authors propose a general approach that can be used for all regimes.

**Theoretical Claims:**

The reviewer checked the correctness of the proofs for theoretical claims for Theorem 3.1. and Proposition 3.5

---

> ### Author Rebuttal · Authors · 2025-04-01
>
> We would like to thank the reviewer for acknowledging the novelty of our work and its mathematical contributions. Our responses to the concerns are provided below.
>
> **Q1) In the preliminaries (line 120-130), the definition of $\sim$ is duplicated ( for probability distribution and for the neighboring sets). After Eq. (11), the authors need to explain the constraint with an example (if $c(x)=x^2$, the constraint limits the variance of additive noise).**
>
> We appreciate the reviewer for pointing out the duplication in the preliminaries and the need for further clarification on the constraint after Eq. (11). We will include an example to clarify the constraint and address the duplication in the final version of the paper.
>
> **Q2) In the contribution parts, the authors mentioned that “the algorithm recovers as special cases noise distributions that are known to be optimal in different regimes, such as the Staircase and Cactus mechanisms.” However, the reviewer cannot find the related experimental results.**
>
> We thank the reviewer for their valuable feedback. Our distribution class is rich enough that it can closely approximate either the Staircase mechanism or the Cactus mechanism. Thus, by the nature of the optimization problem, our resulting distribution will always be at least as good as either mechanism. Specifically, the Cactus mechanism (which is optimal when the number of compositions tends to infinity) is derived by minimizing the KL-divergence, which corresponds to the limiting case of Renyi DP (RDP) as $\alpha$ approaches 1. On the other hand, the RDP of order $\alpha=\infty$ is exactly pure DP, and our optimization will yield the Staircase mechanism as the optimal noise distribution.
>
> As we explained in the introduction, the optimal $\alpha$ value, chosen through the moments accountant formula (used in our algorithm), is closely linked to the number of compositions. This method effectively determines a large $\alpha$ for a single composition with $\delta$ of 0 (pure DP), and an $\alpha$ close to 1 for larger composition scenarios, leading to the Staircase and Cactus distributions as special cases. In the final version, we will include additional figures comparing the noise distributions derived from the settings optimal for Staircase and Cactus, and show how our approach recovers these distributions.
>
> **Q3) In the numerical results, the authors said “a small number of compositions, Laplace is close to the optimal, and the Gaussian is close to the optimal if large number of compositions,” although the optimal one would be star-case and cactus distributions. Can the authors provide experimental results by adding two baselines (stair-case and cactus).**
>
> We appreciate the reviewer’s suggestion to compare against these baselines. While we did not include them in the current experimental results (as our plots primarily focused on settings with a non-zero $\delta$ and moderate composition, where Gaussian and Laplace noise were the better choices and Staircase and Cactus were not optimal), we will include detailed plots in the final version to compare the noise obtained from our method with the Staircase and Cactus baselines.
>
> **Q4, Q5) Are there any additional practical examples regarding moderate composition regimes other than U.S. Census Bureau? Please better introduce the relation between the authors’ work and the machine learning. For example, how can we use moderate competition in Machine Learning field?**
>
> We discussed potential datasets for moderate composition regimes in the rebuttal to Reviewer WDdd (Response to Q2) and provided an example in Response to Q3, demonstrating the improvement on a real-world dataset when using our noise. Due to space constraints, we kindly ask the reviewer to refer to those sections.

---

### Official Review · Reviewer_WDdd · 2025-03-20

**Overall Recommendation:** 2

**Summary:**

The paper proposed a novel framework for optimizing noise distributions for (epsilon, delta)-DP using the Renyi differential privacy formulation. Experiments are shown to showcase the benefits of the approach.

Overall: The paper is easy to follow and the main results are well laid out. The experimental results are not particularly convincing in terms of the practicality of the proposed approach.

**Claims And Evidence:**

Yes

**Essential References Not Discussed:**

They are mostly covered

**Experimental Designs Or Analyses:**

Yes, check below.

**Methods And Evaluation Criteria:**

Yes and no, see below.

**Other Comments Or Suggestions:**

See below

**Other Strengths And Weaknesses:**

Pros:

(i) The setting is highly interesting and it answers the question of how to select the optimal distribution for the specific problem at hand.
(ii) The utilization of the Renyi DP formulation and the subsequent minmax problem with the proposed convex optimization/solver are interesting. In particular, each of these are rigorously proved by the presented Theorems/propositions.
(iii) The experiments show how the optimization finds the best of the Gaussian and Laplacian regimes and in some cases is better than both.

Cons:

(a) The main issue is that the experiments do not convey the power of the proposed approach and except for very narrow regimes, it would be okay to use the Gaussian or the Laplacian setting directly.
(b) The experimental setting is pretty limited and it would be nicer to see further results on ML/DL models referenced in the introduction which would have a substantial number of compositions beyond 10 that are shown here.

**Questions For Authors:**

(1) How do you think this framework would apply to real-world applications? Will there be substantial gains over Gaussian or Laplacian distributions?

**Relation To Broader Scientific Literature:**

See below

**Theoretical Claims:**

Yes, proposition 3.5 and Theorem 3.6

---

> ### Author Rebuttal · Authors · 2025-03-31
>
> We first would like to thank the reviewer for recognizing the novelty of our work and for their constructive comments. Below, we provide a detailed response to their concerns.
>
> **Q1) The main issue is that the experiments do not convey the power of the proposed approach and except for very narrow regimes, it would be okay to use the Gaussian or the Laplacian setting directly.**
>
> We appreciate the reviewer's comment. While Gaussian noise is provably optimal only in the asymptotic setting as sensitivity approaches zero, no such proof exists for Laplace noise in general. Our framework allows us to compute the optimal noise distribution for any setting, and by comparing $(\epsilon, \delta)$ guarantees, we can now see that Gaussian and Laplace noise are near-optimal in certain cases. This insight is a direct outcome of our approach. It should be emphasized that, without our approach, it is not clear when it is best to choose Gaussian versus Laplace. Rather than replacing Gaussian or Laplace noise universally, our method provides a principled way to determine when to use them versus our optimized distribution. Figure 2 comprises a handy guide that tells a practitioner exactly what to do based on their setting.
>
> In the moderate composition regime, highlighted in Figure 2, the improvement achieved by our approach compared to classical distributions is nontrivial. For example, Figure 1 illustrates a setting in which our approach improves the epsilon value by more than 5%. For those on a tight privacy budget, this is significant.
>
> **Q2) The experimental setting is pretty limited and it would be nicer to see further results on ML/DL models referenced in the introduction which would have a substantial number of compositions beyond 10 that are shown here.**
>
> While modern ML often focuses on high-dimensional datasets, where training with DP guarantees requires a high number of compositions, developing predictive algorithms for tabular datasets has always been, and continues to be, a key part of the ML repertoire. In particular, sharing statistics for a restricted number of compositions in a private manner is a key application of DP.
>
> Some tabular datasets come with predefined SQL queries or are used in environments where a limited number of queries is the norm. For example, the U.S. Census Bureau enforces pre-approved aggregate queries to safeguard sensitive data. Similarly, health datasets like NHANES and the Medical Cost Personal Dataset restrict researchers to approved statistical queries, such as calculating the average medical charges for specific age groups. These constraints highlight that in many real-world scenarios, a small number of queries (e.g., 10-20) is both standard and necessary, reinforcing the practical value of optimized noise mechanisms tailored for such settings.
>
> **Q3 ) How do you think this framework would apply to real-world applications? Will there be substantial gains over Gaussian or Laplacian distributions?**
>
> As mentioned above, there are many practical datasets for which a small number of compositions is meaningful. To demonstrate the superior performance of our noise mechanism, **we have an additional experiment that we performed for this review;**  it involves 10 queries on the Medical Cost Personal Dataset. These queries include calculating the average medical charges for specific age groups, total charges by smoker status, average BMI by region, and other similar aggregate statistics. We target the $(\epsilon, \delta) = (2.83, 10^{−6})$ setting. As illustrated in Figure 1 of our paper, Laplace noise achieves this with a standard deviation of 5, outperforming Gaussian noise. To achieve the same $\epsilon$ of 2.83, our noise mechanism requires a **lower standard deviation** of 4.73. To highlight the utility of our mechanism for the abovementioned dataset, we use the metric of MMSE (minimum mean squared error) as follows: we compute the empirical average (over 10 queries, repeated 100k times) of the squared difference of the true query and the clipped+noisy DP query output. The resulting average MMSE for Laplace noise is 24.98, while our optimized noise achieves an MMSE of 22.37, resulting in a 10.4% improvement.
>
> If accepted, we will include these results in the final paper. We hope this example highlights the value of our work and its practical advantages in real-world machine learning applications.

---

> > ### Comment · Reviewer_WDdd · 2025-04-09
> >
> > Thanks to the authors for the detailed rebuttal and it clarified the results further. I have read the responses to mine and other comments and for now will keep my score. It's an interesting result for sure.

---

### Decision · Program_Chairs · 2025-05-01

**Decision:**

Accept (poster)

**Comment:**

This paper makes novel contributions to the development of optimal additive noise distributions for differential privacy. The work is well-presented, appears sound, and may yield nontrivial improvements for real-world applications.

There are some limitations: the gains are relatively modest, there are a number of hyperparameters to be set when implementing the optimization algorithm, and the resulting noise distributions are somewhat opaque and may be hard for practitioners to grasp intuitively. The conceptual novelty is also limited, in the sense that the results don't seem to provide a lot of new fundamental understanding. Still, in settings with a fixed metric and strict budgets, the methods here look promising.

The authors should follow through on the changes promised in response to reviewer questions (comparisons with the staircase mechanism, additional results, etc.), as these will significantly enhance the paper.